# Transcription factor binding sites are frequently under accelerated evolution in primates

Xinru Zhang [1,2,3] ✉, Bohao Fang [4] & Yi-Fei Huang [1,2] ✉

Recent comparative genomic studies have identified many human accelerated elements (HARs) with elevated substitution rates in the human lineage. However, it remains unknown to what extent transcription factor binding sites (TFBSs) are under accelerated evolution in humans and other primates. Here, we introduce two pooling-based phylogenetic methods with dramatically enhanced sensitivity to examine accelerated evolution in TFBSs. Using these new methods, we show that more than 6000 TFBSs annotated in the human genome have experienced accelerated evolution in Hominini, apes, and Old World monkeys. Although these TFBSs individually show relatively weak signals of accelerated evolution, they collectively are more abundant than HARs. Also, we show that accelerated evolution in Pol III binding sites may be driven by lineage-specific positive selection, whereas accelerated evolution in other TFBSs might be driven by nonadaptive evolutionary forces. Finally, the accelerated TFBSs are enriched around developmental genes, suggesting that accelerated evolution in TFBSs may drive the divergence of developmental processes between primates.

During the course of evolution, a subset of genes and regulatory elements may be subject to different pressures of natural selection in distinct species. These genomic elements often have varying substitution rates across species, which may be identified by phylogenetic models with lineage-specific substitution rates[1–10]. Notably, previous studies have revealed a few thousand human accelerated regions (HARs) with dramatically elevated substitution rates in the human lineage compared to other vertebrates[7–15]. A large proportion of HARs are neural enhancers[14–20] and frequently subject to strong positive selection in the human lineage[9–11,21], suggesting that they may contribute to the adaptive evolution of human brain. Also, recent studies show that deleterious mutations in HARs may be associated with neurodevelopmental disorders[22–26], highlighting the key role of HARs in maintaining the integrity of the central nervous system. Thus, characterizing genomic elements under accelerated evolution is of

great importance for understanding the genomic basis of human evolution and disease.

While numerous studies have been conducted to examine accelerated evolution in humans and other species[7–15,27–29], the existing studies may suffer from two critical limitations. First, most of the previous studies have focused on conserved noncoding elements under accelerated evolution. Because a large proportion of noncoding regulatory elements may be subject to frequent evolutionary turnover[30–34], these studies may not be able to characterize accelerated evolution in non-conserved regulatory elements. Second, the previous studies have focused on identifying individual HARs with a genome-wide level of significance. Because of the small amount of alignment data in a single genomic element and the high burden of multiple testing correction associated with a genome-wide scan, these studies may have limited statistical power to detect weakly accelerated

[1]Department of Biology, Pennsylvania State University, University Park, PA 16802, USA. [2]Huck Institutes of the Life Sciences, Pennsylvania State University, University Park, PA 16802, USA. [3]Bioinformatics and Genomics Graduate Program, Pennsylvania State University, University Park, PA 16802, USA. [4]Department of Organismic and Evolutionary Biology and the Museum of Comparative Zoology, Harvard University, Boston, MA 02135, USA. ✉e-mail: xmz5176@psu.edu; yuh371@psu.edu

evolution driven by relaxed purifying selection or weak positive selection. Altogether, it remains unknown to what extent non-conserved genomic elements are subject to weakly accelerated evolution.

Substitutions in transcription factor binding sites (TFBSs) are a main driver of phenotype diversity between species[35–37], implying that TFBSs may also be subject to accelerated evolution. However, to the best of our knowledge, accelerated evolution in TFBSs has not been systematically explored in previous studies, possibly because the majority of TFBSs are not highly conserved across vertebrates[30–33,38,39]. Also, TFBSs might be subject to weaker acceleration compared to conserved elements because the phenotypic effects of mutations are weaker in TFBSs than in conserved elements[40]. Therefore, previous phylogenetic methods dedicated to infer strong signals of accelerated evolution may be underpowered to detect TFBSs under weakly accelerated evolution.

Here, we introduce two novel phylogenetic methods for exploring TFBSs under accelerated evolution. Unlike previous methods that analyze individual elements separately[7–15], our new approaches pool thousands of TFBSs with similar functions together to boost the statistical power to detect weak signals of accelerated evolution. These new methods allow us to rigorously test whether a group of TFBSs as a whole is significantly enriched with accelerated elements, despite that we may lack statistical power to identity individual TFBSs under accelerated evolution due to limited alignment data in a single TFBS. Using these methods, we show that TFBSs of numerous transcription factors are likely to be under accelerated evolution in Hominini, apes, and Old World monkeys. Compared to previously identified HARs, these TFBSs show weaker acceleration but are more abundant genome-wide. Among these accelerated TFBSs, binding sites of DNA-directed RNA polymerase III (Pol III) show the strongest signal of acceleration, which might be driven by strong lineage-specific positive selection on par with HARs. Taken together, accelerated evolution may be a common characteristic of TFBSs in Hominini, apes, and Old World monkeys.

## Results

### Pooling-based phylogenetic inference of accelerated evolution

In the current study, we introduce a novel software application, GroupAcc, which includes two pooling-based phylogenetic approaches with improved statistical power to infer weakly accelerated evolution. The key idea of GroupAcc is to group TFBSs by the bound transcription factor and then examine whether each TFBS group as a whole shows an elevated substitution rate in a lineage of interest. In this study, TFBSs refer to peaks in Chromatin immunoprecipitation-sequencing(ChIP-seq)[41]. By pooling alignment data from a large number of TFBSs, our new methods have significantly higher statistical power to detect weakly accelerated evolution at the group level even if the signals of acceleration are statistically insignificant at the level of individual TFBSs.

In the first method, we utilize a *group-level likelihood ratio test (LRT)* to infer whether a TFBS group as a whole shows an elevated substitution rate in a predefined foreground lineage compared to the other lineage (background lineage) (Fig. 1). To this end, we first fit a reference phylogenetic model to the concatenated alignment of all TFBSs, where we estimate the branch lengths of a phylogenetic tree, the gamma shape parameter for rate variation among nucleotide sites[42], and the parameters of the general time reversible substitution model[43]. Assuming that the majority of TFBSs may not be under accelerated evolution, the reference phylogenetic model may represent the overall pattern of sequence evolution in TFBSs when accelerated evolution is absent. Given the reference phylogenetic tree, we then fit the group-level LRT to the concatenated alignment of a TFBS group, where we estimate two scaling factors, $r_1$ and $r_2$, for the foreground and the background branches, respectively. We interpret $r_1$

and $r_2$ as the relative substitution rates of the TFBS group in the foreground and background lineages throughout this study. We assume that $r_1 = r_2$ in the null model ($H_0$), indicating that the TFBS group has evolved at a constant rate across all lineages. Conversely, we assume that $r_1 \neq r_2$ in the alternative model ($H_a$), indicating that the TFBS group has evolved at different substitution rates between the foreground and background lineages. Because the TFBS group may consist of hundreds of TFBSs, we assume that the likelihood ratio statistic of the group-level LRT asymptotically follows a chi-square distribution with one degree of freedom. If the null model is rejected in the group-level LRT and $r_1 > r_2$, we consider that the TFBS group as a whole may be subject to accelerated evolution.

In the second method, we use a *phylogenetics-based mixture model* to estimate the proportion of accelerated TFBSs in a TFBS group (Fig. 1). To this end, we first perform an element-level LRT to infer evidence for accelerated evolution in individual TFBSs given that $H_0$ is rejected in the group-level LRT. The element-level LRT is similar to the group-level LRT but is applied to the alignments of individual TFBSs rather than to the concatenated alignment of the TFBS group. Given the likelihood ratio statistics from the element-level LRT, we then calculate empirical *p*-values for individual TFBSs using parametric bootstrapping. Unlike the chi-square distribution in the group-level LRT, the parametric bootstrapping procedure provides accurate *p*-values even when there is a small amount of alignment data per test[9]. Finally, we estimate the proportion of accelerated TFBSs in the TFBS group by fitting a beta-uniform mixture model to the distribution of *p*-values[44]. The beta-uniform mixture model allows us to estimate an upper bound of the proportion of TFBSs generated from $H_0$ ($\hat{\pi}_{ub}$). We consider $1 - \hat{\pi}_{ub}$ as a conservative estimate (lower bound) of the proportion of accelerated TFBSs.

### GroupAcc is able to identify weakly accelerated evolution in synthetic data

To verify the power of GroupAcc to infer weakly accelerated evolution in TFBS groups, we conducted simulations under various lineage-specific evolutionary dynamics. In the first scenario, we assumed that all the binding sites in one group are under accelerated evolution in a specific lineage. The second scenario considered the heterogeneity of evolutionary patterns in each single binding site: only parts of each binding site (for example, motif) undergo accelerated evolution in a specific lineage. The third scenario considered the heterogeneity of evolutionary dynamics in groups of binding sites: only certain numbers of binding sites in one group undergo accelerated evolution in a specific lineage, while the other binding sites do not undergo accelerated evolution. Under each scenario, we verified the ability of the group-level LRT method to detect accelerated evolution in a specific lineage and estimate the fold of increase in substitution rate ($r_1/r_2$). We also compared the performance of the phylogenetics-based mixture model and traditional element-level LRT in estimating the number of elements under accelerated evolution in a given lineage.

In each scenario, eight cases were generated in which different lineages of primates were under accelerated evolution: (1) only human, (2) subtree of Hominini (human, chimp), (3) subtree of human, chimp, and gorilla, (4) subtree of Great apes (chimp, gorilla, orangutan) and human, (5) only chimp, (6) only gorilla, (7) only orangutan, (8) only macaque. For each case, we simulated alignments of 10,000 binding sites, each at the length of 200 bp based on the reference model plus those assumptions. We also simulated alignments of different numbers of binding sites (1000) and different lengths of each alignment (100 bp) to test the performance of the model in different settings. Both weak and strong accelerated evolution were taken into consideration: the fold of increase in substitution rate in foreground lineage ($r_1/r_2$) varied from 1.2 to 5.

Under the first scenario, all the 200 bp binding sites in one group were assumed to be under accelerated evolution in a defined lineage as

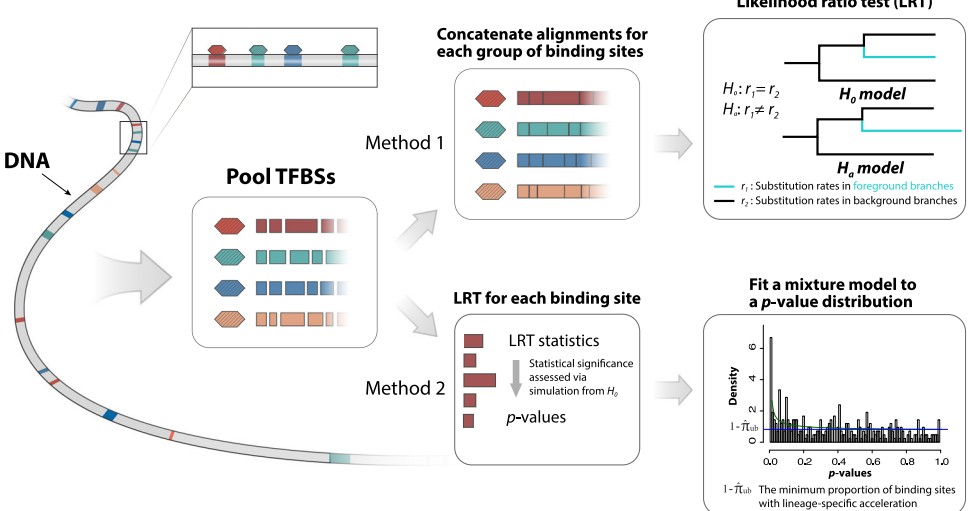

**Fig. 1 | Pooling-based phylogenetic methods for inferring accelerated evolution in TFBSs.** In the first method, we fit the group-level LRT to the concatenated alignment of TFBSs bound by the same transcription factor, which allows us to examine whether the group of TFBSs as a whole evolved at an elevated substitution rate in the foreground lineage compared to the background lineage. In the second method, we fit the element-level LRT to the alignment of each individual TFBS, which provides an element-level $p$-value. Then, we fit a beta-uniform mixture model to the distribution of $p$-values in each TFBS group to estimate the proportion of accelerated TFBSs. Colored rectangles and hexagons represent TFBSs and transcription factors, respectively. $r_1$ and $r_2$ represent relative substitution rates in the foreground and background lineages, respectively.

each case (1–8) showed, for example, in case 1, all the 200 bp binding sites would be under accelerated evolution in only human. With foreground lineage matching with the accelerated lineage in each case, the group-level LRT method was able to tell the presence of accelerated evolution at the group level and accurately estimate the fold of increase in substitution rate in foreground lineages ($r_1/r_2$), even given weak accelerated evolution when the fold of increase in substitution rates is only slightly larger than 1 (Fig. 2a). The GroupAcc model performed better than element-level LRT in estimating the number of elements under accelerated evolution (Fig. 2b). We also tested if the model could detect accelerated evolution in a tip if a subtree containing the tip is under accelerated evolution (Fig. 3). In cases (1), (2), (3) and (4), when accelerated evolution happened in lineages such as human or subtrees containing human, taking human as foreground lineage, GroupAcc methods were able to identify the presence of accelerated evolution in human and estimate the number of elements under accelerated evolution in human with higher accuracy compared to traditional element-level LRT method (Fig. 3). In cases (5), (6), (7) and (8), when accelerated evolution occurred in lineages other than human, the GroupAcc methods were able to identify the fact that human is not undergoing accelerated evolution (Fig. 3).

We validated the ability of our methods to identify lineage-specific acceleration when only part of the TFBS is under accelerated evolution from simulation scenario 2. We generated 10,000 200 bp alignments standing for elements. Each alignment was composed of $200 \times L$ bp generated with a scaled tree (with substitution rate increase) and $200(1-L)$ bp generated from an unscaled tree (without substitution rate increase). Given that $L = 0.1, 0.2, 0.5, 0.8$. the group-level LRT was able to identify the presence of accelerated evolution, even under weak acceleration when the fold of substitution rate increase in foreground lineage was only 1.2 (Fig. 4a). The GroupAcc method outperformed the element-level LRT method in estimating the number of elements under accelerated evolution (Fig. 4b). Therefore, under situations with the heterogeneity of evolutionary patterns in each single binding site, our pooling based methods were able to identify lineage-specific accelerated evolution with a uniform scaling of substitution rates on the foreground lineages across the whole binding sites.

Under the third scenario, a specific proportion M of binding sites ($M = 0.1, 0.2, 0.5, 0.8$) in a group were under accelerated evolution. This scenario considered the heterogeneity of evolutionary dynamics in multiple binding sites of one transcription factor. We found group-level LRT method was able to tell the presence of accelerated evolution at the group level and estimate the fold of increase in the substitution rate of foreground lineages, even when the fold of increase in substitution rates of foreground lineage was slightly larger than 1 (Fig. 5). The GroupAcc model performed better than element-level LRT in estimating the number of elements under accelerated evolution (Fig. 5).

**Numerous TFBS groups show evidence for accelerated evolution**
Using the group-level LRT, we examined accelerated evolution in 4,380,444 TFBSs of 161 transcription factors identified by ChIP-seq experiments in the ENCODE Project[45]. We tested whether each group of TFBSs bound by the same transcription factor had an elevated substitution rate in the human lineage. We used Multiz genome alignments of ten primate species[46] and defined the human lineage after the divergence of chimpanzees and humans as the foreground lineage. Unlike previous studies of HARs[7–15], we did not include non-primate vertebrates to mitigate the impact of the evolutionary turnover of TFBSs on our analysis[30–33,38,39]. After Bonferroni correction, we observed that 15 TFBS groups had significantly different substitution rates between the foreground and background lineages (Supplementary Data 1), which all showed elevated substitution rates in humans compared to other primates ($r_1 > r_2$).

TFBS groups with elevated substitution rates in humans could be either directly under accelerated evolution or merely overlapping with other accelerated TFBS groups. To identify TFBS groups directly under accelerated evolution, we sought to partition the binding sites of the 15 TFBS groups with elevated substitution rates into non-overlapping, biologically interpretable TFBS groups. Because BDP1, BRF1, and POLR3G are components of the Pol III transcription machinery[47], we defined a new TFBS group, Pol III binding, consisting of genomic regions bound by at least two of the three transcription factors. Similarly, since POU5F1 and NANOG can interact with each other to form a protein complex[48,49], we defined another TFBS group, POU5F1-

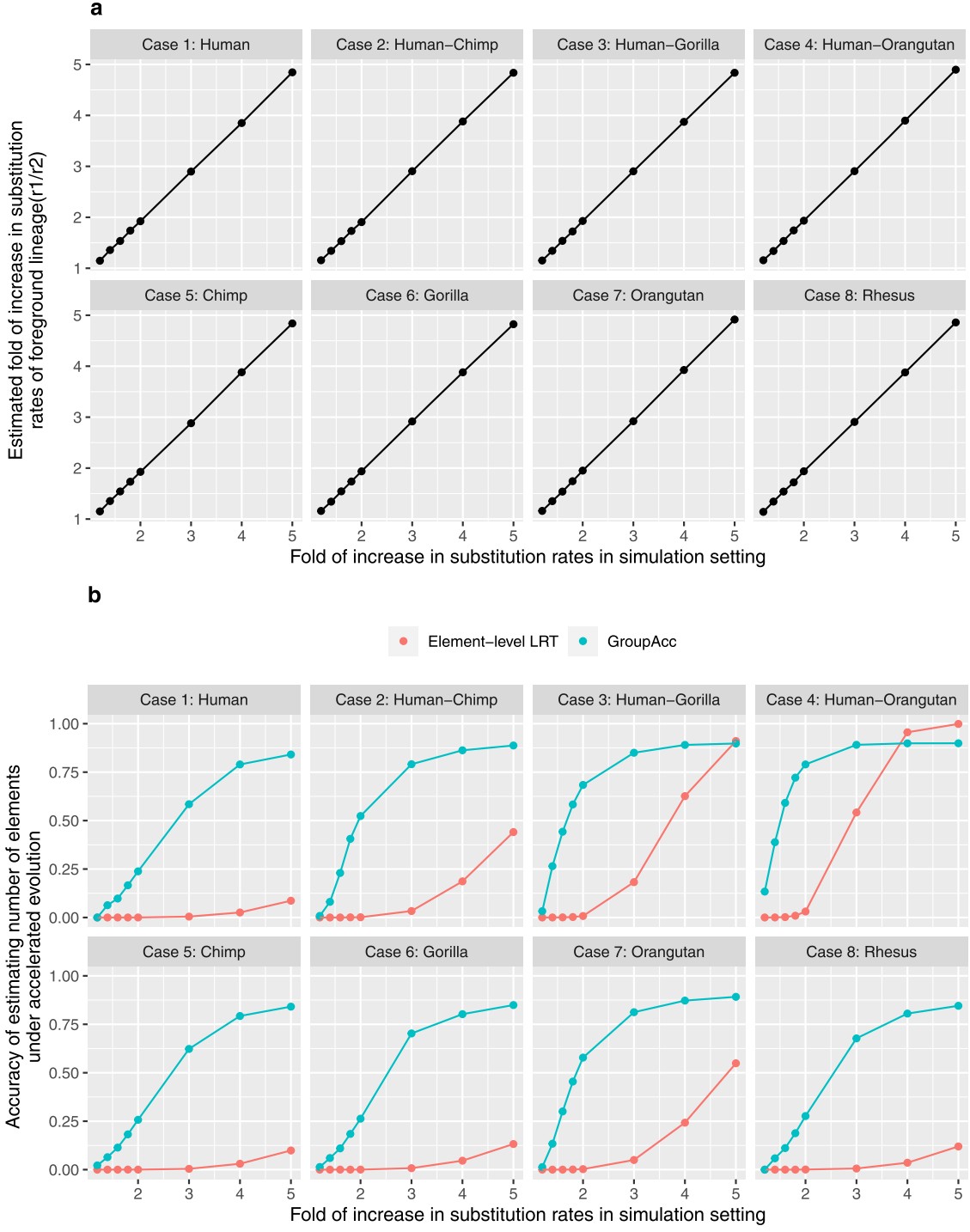

**Fig. 2 | Simulation results of scenario 1 with foreground lineage matching the accelerated lineage in each case (1–8). a** *X*-axis shows the scaling factor of foreground lineage branch length in simulation setting, which is the real fold of increase in the substitution rate of the foreground lineage. *Y*-axis shows the fold of increase in the substitution rate of the foreground lineage estimated from group-level LRT. **b** Comparison of accuracy estimating the number of elements under accelerated evolution between GroupAcc and element-level LRT method. Blue curves are the accuracy of GroupAcc. Red curves are the accuracy of element-level LRT.

NANOG binding, consisting of genomic regions bound by both of the two transcription factors.

Then, we removed all binding sites overlapping more than one TFBS group, resulting in 17 non-overlapping TFBS groups. We applied the group-level LRT again to these non-overlapping TFBS groups. After Bonferroni correction, seven non-overlapping TFBS groups showed significantly elevated substitution rates in the human lineage (Fig. 6; Supplementary Table 1). These non-overlapping TFBS groups included Pol III binding, POU5F1-NANOG binding, BDP1, FOXP2, POU5F1,

NANOG, and NRF1. Compared to previously identified HARs, the seven non-overlapping TFBS groups showed weaker acceleration as evidenced by their smaller increases in substitution rates in the human lineage (Fig. 6). We focused on the seven non-overlapping TFBS groups with evidence for weakly accelerated evolution in downstream analysis.

**Accelerated evolution in TFBSs may not be human specific**
A recent study showed that many HARs may also undergo accelerated evolution in other apes[15]. Based on the simulation of

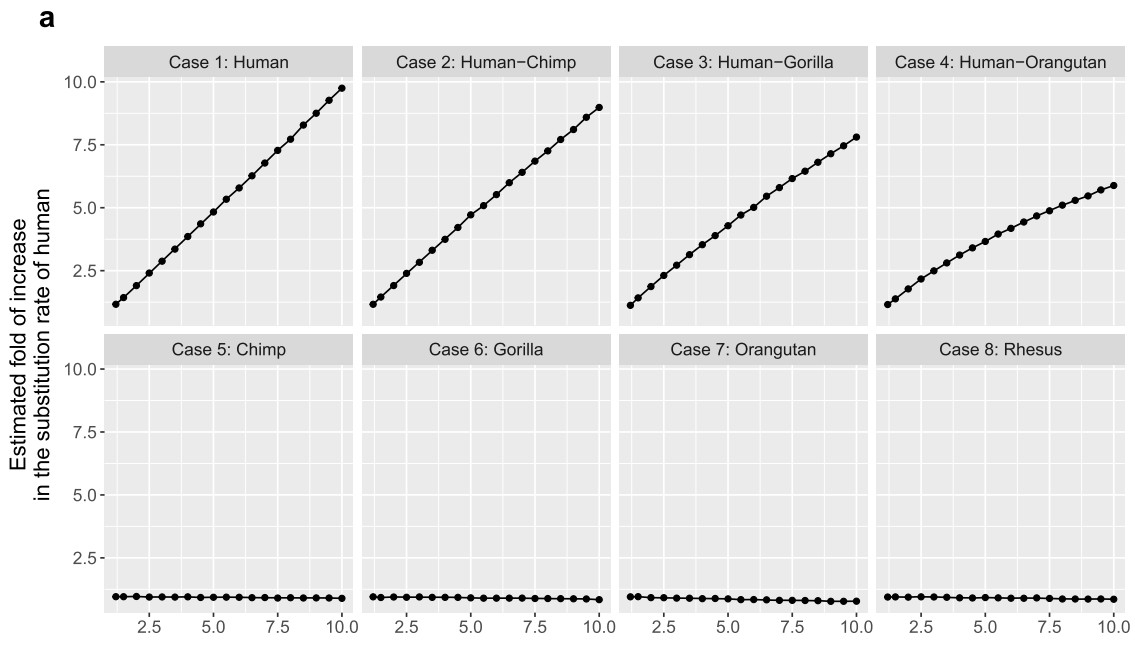

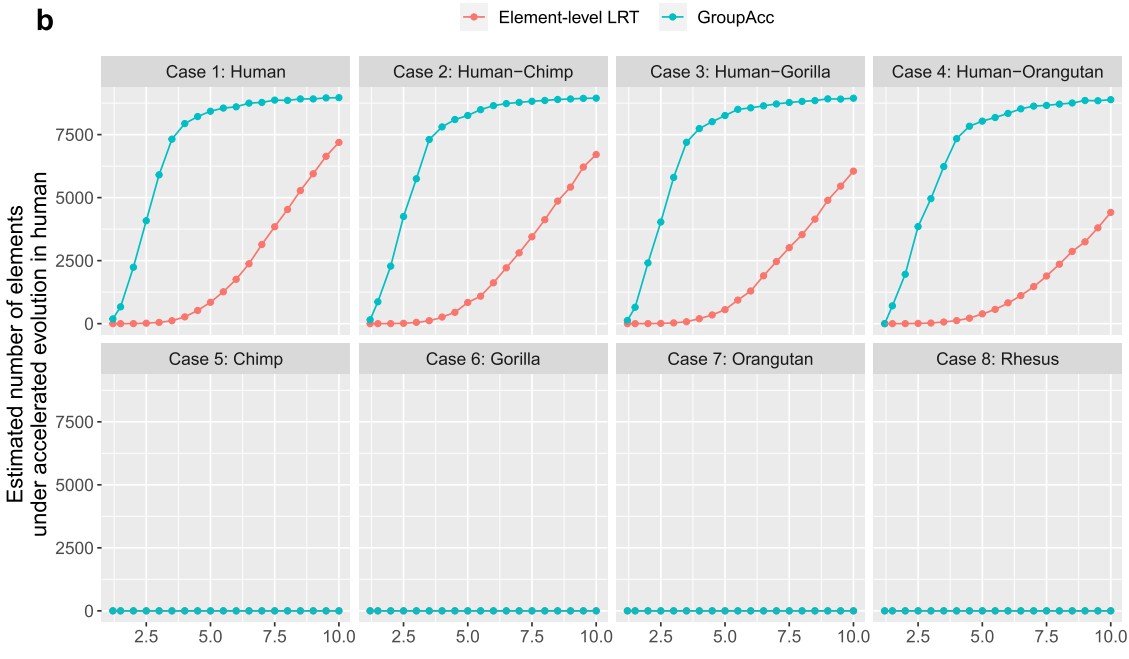

**Fig. 3 | Simulation results of scenario 1 using human as foreground lineage. a** *X*-axis shows the scaling factor of accelerated lineage branch length in simulation setting, which is the real fold of increase in the substitution rate of the accelerated lineage. *Y*-axis shows the fold of increase in the substitution rate of human estimated from group-level LRT. **b** Comparison of the estimated number of elements under accelerated evolution in human between GroupAcc and element-level LRT method. Blue curves are the estimates of GroupAcc. Red curves are the estimates of element-level LRT.

scenario 1 (Fig. 3), we found that the GroupAcc methods were able to tell the presence of accelerated evolution in human lineage at the group level when accelerated evolution occurred in any subtrees containing human. To characterize when acceleration occurred during the evolution of TFBSs, we employed a model comparison approach to search for lineages with elevated substitution rates. Specifically, we evaluated the goodness-of-fit of seven phylogenetic models with different foreground lineages, denoted as M1 to M7 (Fig. 7a). All these models were based on $H_a$ in

the group-level LRT (Fig. 1), and the foreground lineages associated with these models corresponded to all the monophyletic clades that included the human lineage (Fig. 7a). These models effectively assumed that the change of substitution rate occurred at most once during the evolution of a TFBS group, which was designed to explore the most parsimonious explanations of accelerated evolution and to limit the number of tested foreground lineages. We used the Bayesian information criterion (BIC) as a measure of goodness-of-fit of these models.

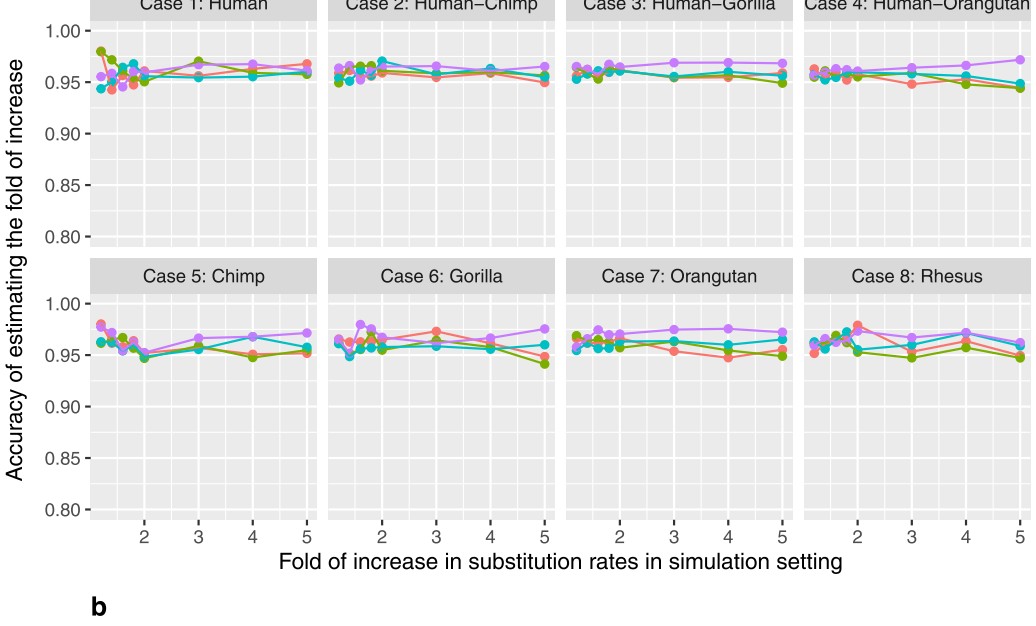

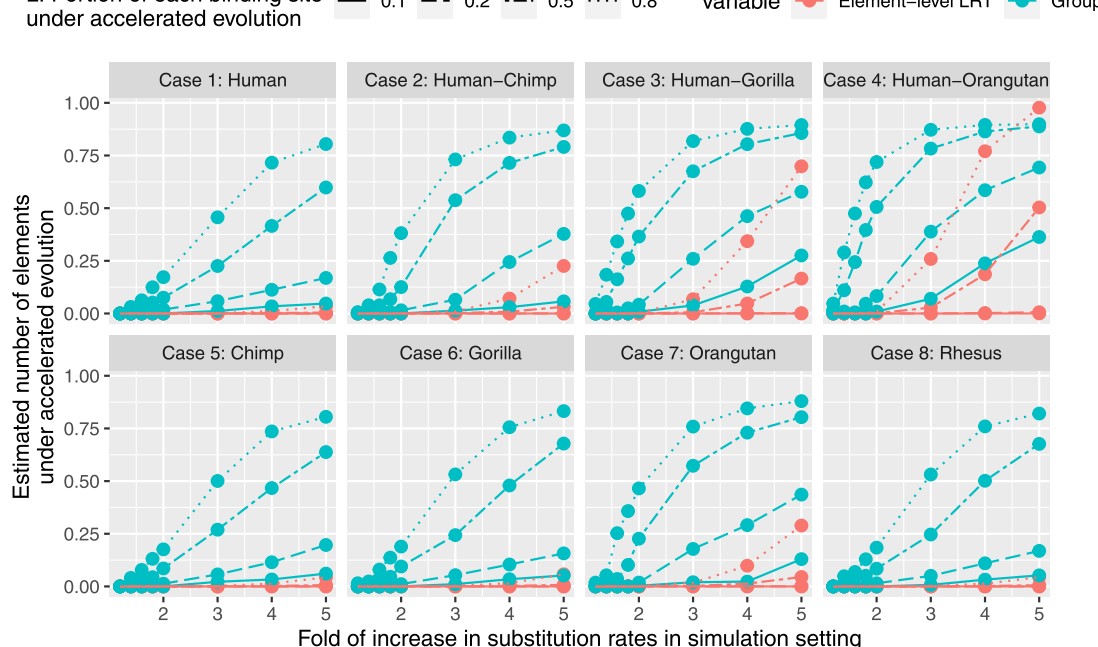

**Fig. 4 | Simulation results of scenario 2. a** Accuracy of GroupAcc in estimating the fold of increase in the substitution rate of foreground lineage given different portions of each binding site under accelerated evolution. *X*-axis shows the scaling factor of accelerated lineage branch length in simulation setting, which is the real fold of increase in the substitution rate of accelerated lineage. Y-axis shows the accuracy of estimating the fold of increase in the substitution rate of the accelerated lineage. The weighted estimate of the fold of increase in the substitution rate of foreground lineage across the whole group of binding sites is calculated by $(L \times \hat{r_1}/\hat{r_2} + 1 - L)$. The accuracy of estimating the fold of increase is calculated as $\frac{\text{The weighted estimate of the fold of increase in the substitution rate of foreground lineage}}{r_1/r_2 \text{ in the simulation setting}}$. **b** Comparison of performance estimating the number of elements under accelerated evolution between GroupAcc and element-level LRT method. Blue curves are the estimated numbers of GroupAcc. Red curves are the estimated numbers of element-level LRT.

Although the seven accelerated TFBS groups were originally detected using humans as the foreground lineage, our model comparison analysis showed that accelerated evolution may not be human specific (Fig. 7b and Supplementary Table 2). Specifically, for binding sites of Pol III, BDP1, and NRF1, a model with both apes

and Old World monkeys as the foreground lineage (M5) showed the best goodness-of-fit. Similarly, for binding sites of FOXP2 and NANOG, a model with apes as the foreground (M4) showed the best goodness-of-fit. Moreover, for POU5F1-NANOG and POU5F1 binding sites, a model with Hominini as the foreground (M2)

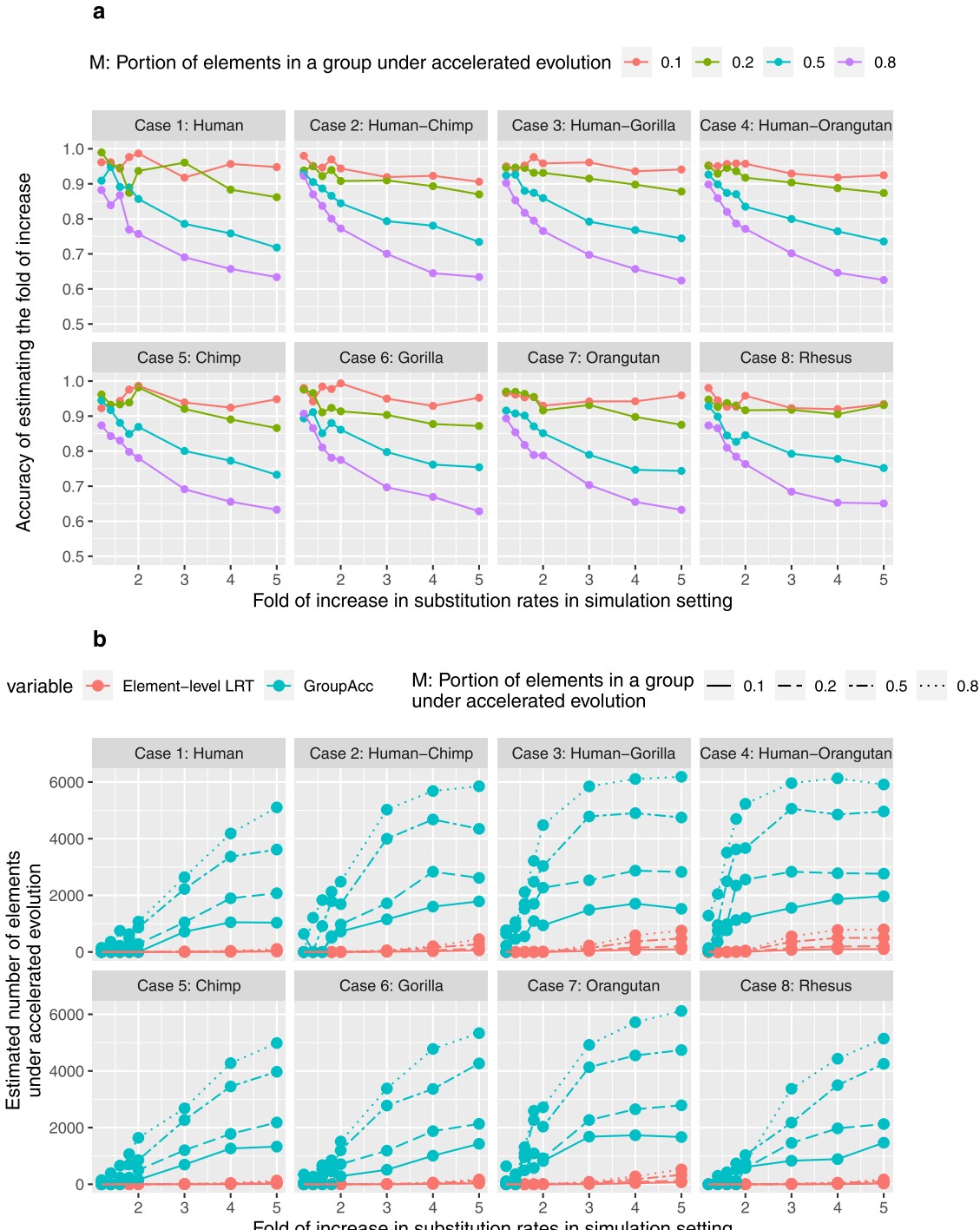

**Fig. 5 | Simulation results of scenario 3. a** Accuracy of GroupAcc in estimating the fold of increase in the substitution rate of foreground lineage given different portions of elements in a group under accelerated evolution. *X*-axis shows the scaling factor of accelerated lineage branch length in simulation setting, which is the real fold of increase in the substitution rate of accelerated lineage. *Y*-axis shows the accuracy of estimating the fold of increase in the substitution rate of the accelerated lineage. The weighted estimate of the fold of increase in the substitution rate of foreground lineage across the whole group of binding sites is calculated by $(M \times \hat{r}_1/\hat{r}_2 + 1 - M)$. The accuracy of estimating the fold of increase is calculated as $\dfrac{\text{The weighted estimate of the fold of increase in the substitution rate of foreground lineage}}{r_1/r_2 \text{ in the simulation setting}}$. **b** Comparison of performance estimating the number of elements under accelerated evolution between GroupAcc and element-level LRT method. Blue curves are the estimated numbers of GroupAcc. Red curves are the estimated numbers of element-level LRT.

showed the best goodness-of-fit. Altogether, the acceleration of TFBS evolution might be driven by changes of selection pressure in Hominini, apes, and Old World monkey and, thus, might contribute to phenotypic differences between these species and other primates.

**More than 6000 TFBSs may be under accelerated evolution**
In this section, we sought to infer the total number of TFBSs under accelerated evolution. While the group-level LRT can examine whether a TFBS group as a whole was under accelerated evolution, it could not estimate the number of accelerated TFBSs in the TFBS group. Also,

because the signal of acceleration might be weak in TFBSs (Fig. 6), previous phylogenetic models could not be used to estimate this number either[7–15]. To address this problem, we utilized the phylogenetics-based mixture method to estimate the proportion of accelerated TFBSs from the distribution of *p*-values associated with individual TFBSs in the same group (Fig. 1).

We observed that 78% of Pol III binding sites were under accelerated evolution (Table 1 and Supplementary Table 3), which translates to approximately 222 accelerated Pol III binding sites. Also, 20 and 25%

of binding sites of BDP1 and NRF1 were under accelerated evolution in Old World monkeys and apes, which translates to approximately 90 and 466 accelerated TFBSs, respectively. Approximately 25% of TFBSs of FOXP2 and NANOG were under accelerated evolution in apes, suggesting about 5000 binding sites in these TFBS groups were accelerated elements. Furthermore, approximately 8% of TFBSs of POU5F1 and POU5F1-NANOG were under accelerated evolution in Hominini, indicating that about 300 binding sites of the two groups were accelerated in the clade consisting of humans and chimpanzees. In total, more than 6000 TFBSs spanning 1573kb were under accelerated evolution in Hominini, apes, and Old World monkeys (Table 1), which is more than the 3098 known HARs spanning 720 kb (see "Methods").

## Positive selection may drive accelerated evolution in Pol III binding sites

The acceleration of TFBS evolution could be due to either positive selection or relaxed purifying selection in the foreground lineage. To examine whether positive selection is a driver of accelerated evolution in TFBSs and estimate the selection pressure in TFBSs, we employed the INSIGHT model[50–52] to infer the strength of positive selection and selection pressure on the seven accelerated TFBS groups in the human lineage. Similar to the McDonald-Kreitman test[53,54], INSIGHT incorporates divergence and polymorphism data to infer positive selection on a set of predefined genomic elements. We fit the INSIGHT model to the binding sites of each TFBS group, which provided an estimate of $D_p$, that is, the expected number of adaptive substitutions per kilobase in the human lineage, as well as an estimate of $\rho$ which is the fraction of sites under selection within functional elements.

We observed that Pol III binding sites were subject to strong positive selection in the human lineage, because $D_p$ of Pol III binding sites was significantly higher than 0 and was comparable to that of previously identified HARs (Fig. 8; Supplementary Table 4). By downsampling the 286 Pol III binding sites to 200 or 240 binding sites, we verified that the positive selection could still be detected in Pol III binding sites. In other TFBS groups, $D_p$ was not significantly different from 0, indicating that positive selection might not be the driving force of accelerated evolution in these TFBS groups. Each of the seven TFBS groups were inferred to have a smaller fraction of sites under selection

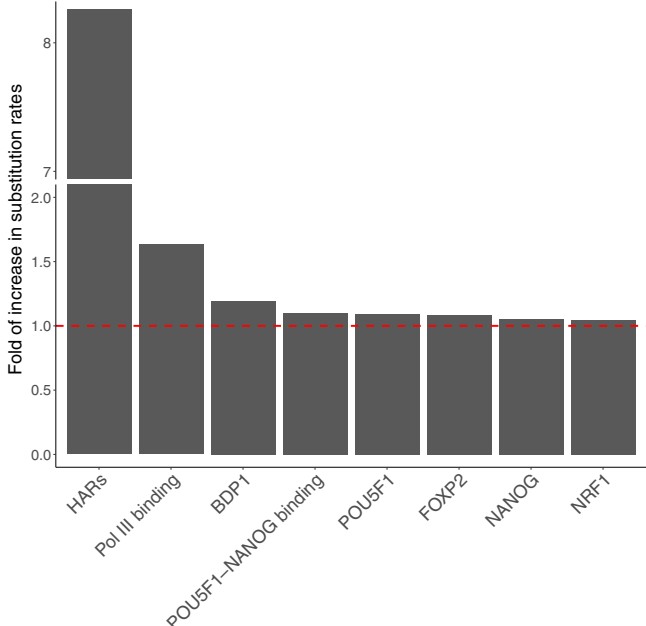

**Fig. 6 | Non-overlapping TFBS groups under accelerated evolution in the human genome.** The fold of increase in substitution rate is defined as $r_1/r_2$, where $r_1$ and $r_2$ are the relative substitution rates of a TFBS group in the human lineage and in other primates, respectively.

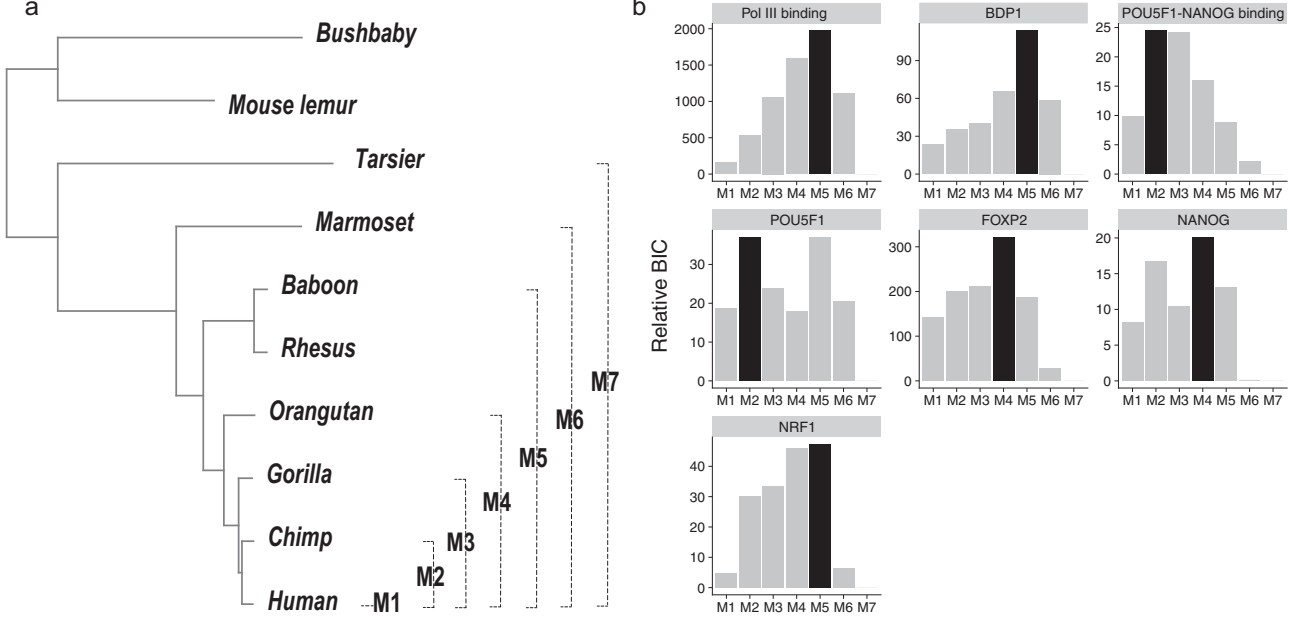

**Fig. 7 | Lineages associated with accelerated evolution in TFBS groups. a** The seven foreground lineages examined in the model comparison analysis. **b** Model fit with different foreground lineages. The black bars indicate the best-fit foreground lineages.

**Table 1 | Numbers of accelerated TFBSs estimated by the phylogenetic mixture model**

| TFBS group | Proportion of accelerated elements $(1 - \hat{\pi}_{ub})$ | Number of elements | Number of accelerated elements | Lineage with accelerated evolution | Selection coefficient $\rho$ | Gene conversion disparity $B$ |
|---|---|---|---|---|---|---|
| Pol III binding | 0.78 | 286 | 222.30 | OWM & ape (M5) | 0.03 | 0.21 |
| BDP1 | 0.20 | 439 | 89.75 | OWM & ape (M5) | 0.02 | 0.24 |
| POU5F1-NANOG binding | 0.08 | 1341 | 109.90 | Hominini (M2) | 0.19 | 0.06 |
| POU5F1 | 0.10 | 2040 | 204 | Hominini (M2) | 0.09 | 0 |
| FOXP2 | 0.27 | 15881 | 4264.92 | Ape (M4) | 0.20 | 0 |
| NANOG | 0.26 | 2952 | 771.21 | Ape (M4) | 0.23 | 0 |
| NRF1 | 0.25 | 1856 | 466.34 | OWM & ape (M5) | 0.07 | 2.0 |

in human $\rho$ (Table 1) than the collection of 161 TFBS groups ($\rho = 0.76$). The reduced values of $\rho$ implied weaker selection constraints in the seven TFBS groups. Applying phastBias[55] to the seven TFBS groups, we observed GC-biased gene conversion in NRF1 binding sites (Table 1).

### The accelerated TFBSs are enriched around developmental genes

To identify the major functions represented by the top accelerated binding sites in the seven TFBS groups, we utilized Genomic Regions Enrichment of Annotations Tool (GREAT) to first find the potential target genes by predicting both proximal and distal binding events, and then analyzed the functional significance of those top accelerated binding sites by applying GO enrichment test and pathway enrichment analysis to their potential target genes with background gene lists composed of all the genes associated with the whole TFBS group[56–58].

We extracted the significant binding sites in each of the seven groups from the phylogenetics-based mixture model and defined

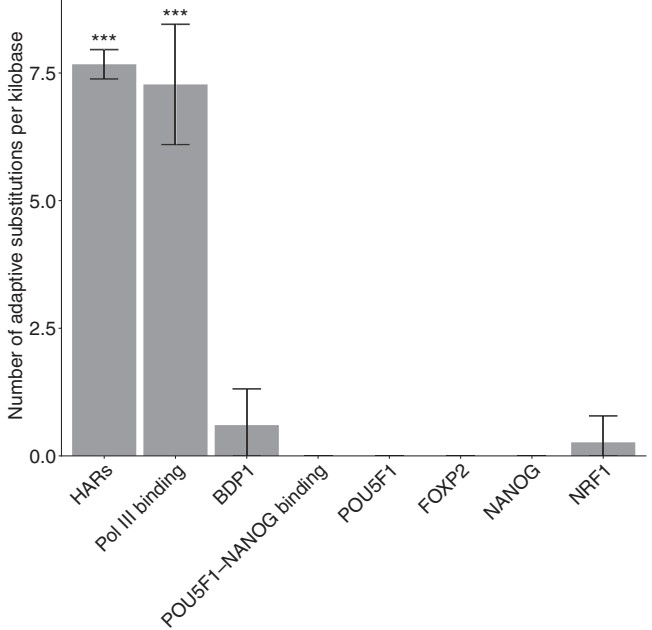

**Fig. 8 | Positive selection on accelerated TFBS groups in the human lineage.** The numbers of adaptive substitutions per kilobase $D_p$ and the standard errors $SE(D_p)$ are estimated by INSIGHT[50–52] (Supplementary Table 3). Error bars are centered at the MLE of $D_p$ estimates and indicate two-fold standard errors in each direction. Total number of genetic elements: $n = 27,893$ (HARs: $n = 3098$, Pol III binding: $n = 286$, BDP1 binding sites: $n = 439$, POU5F1-NANOG binding: $n = 1341$, POU5F1 binding sites: $n = 2040$, FOXP2 binding sites: $n = 15,881$, NANOG binding sites: $n = 2952$, NRF1 binding sites: $n = 1856$). $P$-values were estimated from the one-sided Wald test to compare if $D_p$ is greater than 0. Estimates of $D_p$ found to be significantly greater than 0 are highlighted with stars, ***$p < 0.001$.

them as the top accelerated binding sites. GREAT identified 2611 potential target genes for the top accelerated binding sites of FOXP2, 662 genes for the top accelerated binding sites of NANOG, 390 genes for the top accelerated binding sites of NRF1, 222 genes for the top accelerated binding sites of POU5F1, 163 genes for the top accelerated binding sites shared by POU5F1 and NANOG, 104 genes for the top accelerated binding sites of BDP1 and 143 genes for the top accelerated binding sites shared by Pol III TFs. Using default settings in GREAT, we built seven background gene lists for seven TFBS groups, respectively containing 9896 potential target genes for FOXP2 binding sites, 3745 genes for NANOG binding sites, 2931 genes for NRF1 binding sites, 1976 genes for POU5F1-NANOG binding sites and 478 potential target genes for POU5F1 binding sites.

After removing the redundant GO terms with high semantic similarity (0.7) and performing Bonferroni correction on the GO enrichment results, we found FOXP2 top accelerated TFBSs were associated with genes functioning in artery development and regulation of transforming growth factor signaling pathway. The concatenation of top accelerated binding sites in seven TFBS groups were associated with genes playing roles in development and cell proliferation processes (Fig. 9; Supplementary Data 2).

In the genes associated with other accelerated TFBS groups, no pathways or biological terms were found to be significant after correction. Benjamini–Hochberg correction has been applied to the GO enrichment results. After Benjamini–Hochberg correction, developmental process terms were also enriched for the genes nearby the top accelerated binding sites among the seven groups (Supplementary Data 2).

### Accelerated evolution in primates' ChIP-seq peaks

To investigate the accelerated evolution in primates, we applied the GroupAcc method to datasets that were not human-centric, including non-human primates' ChIP-seq peaks. Vermunt et al.[59] identified histone H3 lysine 27 acetylation (H3K27ac) enriched regions in human, chimpanzee and rhesus macaque brain. The H3K27ac enriched regions were predicted to be active cis-regulatory elements(CREs), We applied the group-level LRT method to the predicted CREs in human, chimpanzee and rhesus macaque brain. Results revealed a slight increase in substitution rates of human and chimpanzee lineage in CREs of human and chimpanzee brain, compared to the fold of increase in substitution rate of rhesus macaque lineage in CREs of rhesus macaque brain.

Villar et al.[60] identified trimethylated lysine 4 of histone H3 (H3K4me3) enriched regions and H3K27ac enriched regions in liver of 20 mammals including human and rhesus macaque. The regions were classified into active gene promoters and enhancers. Enhancers were identified by regions only enriched for H3K27ac, while promoters defined as regions containing both H3K27ac and H3K4me3. We applied the group-LRT method to the promoters and enhancers in human and rhesus macaque. Results showed that enhancers tended to evolve faster than promoters in both species.

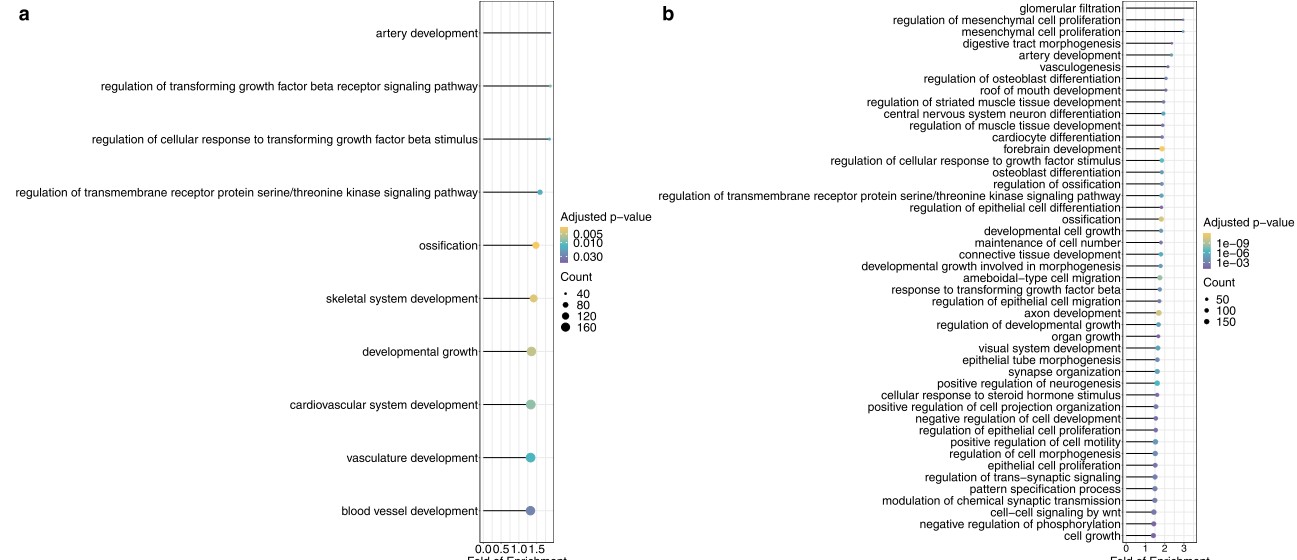

**Fig. 9 | Gene ontology analysis of the genes associated with top accelerated binding sites.** The dot plots show the significant GO terms after Bonferroni correction for biological process of (**a**) genes associated with top accelerated binding sites of FOXP2 (**b**) genes associated with top accelerated binding sites of all seven TFBS groups. The size of circle represents the number of genes associated with top accelerated binding sites affiliated with the specific GO terms. The color of circle represents the Bonferroni-corrected *p*-values.

To identify accelerated evolution in tissue-specific genetic regulatory elements, we applied the GroupAcc method to the most abundant TFBS group: CTCF binding sites. We included 80074 CTCF binding sites across 29 tissues or cell types[61]. We found lower leg skin and tibial nerve CTCF binding sites undergo weak accelerated evolution in human.

## Discussion

In the current study, we present two pooling-based methods to infer genomic elements under accelerated evolution. Unlike previous methods that focus on analyzing individual elements[7–15], our new methods group hundreds of genomic elements with similar biological functions to increase the sample size per test and reduce the multiple testing burden. Thus, our methods may have higher sensitivity to detect weak signals of accelerated evolution. To the best of our knowledge, our methods are the first statistical framework dedicated to inferring weakly accelerated evolution in non-coding regions.

Using the group-level LRT, we identify seven groups of non-overlapping TFBSs with significant evidence for accelerated evolution (Fig. 6). The model comparison analysis suggests that these TFBS groups may be under accelerated evolution not only in humans but also in other primate species (Fig. 7). In agreement with our finding, a recent study of HARs has shown that many HARs may also be subject to accelerated evolution in other ape species[15]. Therefore, accelerated evolution of regulatory elements may be a shared characteristic of primates rather than specific to the human lineage.

Among the seven groups of accelerated TFBSs, we show that Pol III binding sites may be subject to positive selection in the human lineage but find no evidence for positive selection in other accelerated TFBS groups (Fig. 8). In contrast, more than half of HARs may be subject to positive selection in the human lineage[21], suggesting that positive selection may be the main driving force of accelerated evolution in HARs. Because previous studies of HARs have focused on identifying individual genomic elements with extremely high substitution rates in the human lineage, the higher frequency of detecting positive selection in HARs could partially reflect the lower power of previous methods in discovering weakly accelerated elements driven by evolutionary forces other than positive selection.

Although accelerated evolution is much weaker in Pol III binding sites than in HARs (Fig. 6), Pol III binding sites are subject to strong positive selection in the human lineage, on par with HARs (Fig. 8). Because HARs are highly conserved across species, they may have very low substitution rates in non-human primates, which in turn enhances the signals of accelerated evolution. In contrast, Pol III binding sites may not be highly conserved across species, resulting in a weaker signal of accelerated evolution despite strong positive selection in the foreground lineage. Taken together, weak signals of accelerated evolution may not always imply weak positive selection in the foreground lineage.

Other than lineage-specific positive selection, we find that non-adaptive evolutionary forces, such as relaxed purifying selection and GC-biased gene conversion, may drive the accelerated evolution of TFBSs[21,55]. GC-biased gene conversion has been found in NRF1 binding sites but not in other accelerated TFBS groups. The seven groups of accelerated TFBSs have reduced values of the fraction of sites under selection ρ in human comparing to the collection of 161 groups of TFBSs. Overall, the seven groups of TFBSs are under weaker selection constraints than other TFBSs. The widespread nonadaptive evolutionary forces do not indicate the lack of functional importance of those accelerated regions.

Notably, the seven groups of accelerated TFBSs may play key roles in developmental processes. First, recent studies suggest that disruptive mutations in subunits of Pol III, such as *POLR3A*, *POLR3B* and *BRF1*, may be associated with neurodevelopmental disorders[62–64]. Therefore, accelerated evolution in Pol III binding sites might be associated with the adaptive evolution of the central nervous system in apes and Old World monkeys (Fig. 7). Second, POU5F1 and NANOG are transcription factors necessary to the pluripotency and self-renewal of embryonic stem cells[65–67]. The colocalization of POU5F1 and NANOG in regulatory elements, referred to as POU5F1-NANOG binding in the current study, might trigger zygotic gene activation in vertebrates[48,68–71]. Third, FOXP2 is a highly conserved vertebrate protein with high expression in the central nervous systems during embryogenesis, and detrimental mutations in the *FOXP2* gene may cause impaired speech development in humans[72–74]. Also, previous studies have shown that the protein sequence and expression of *FOXP2* could be subject to accelerated evolution in humans[75–77], echolocating

bats[78], and vocal learning birds[79]. Finally, NRF1 has been found to regulate the expression of *GABRB1*, a gene associated with neurological and neuropsychiatric disorders[80,81]. To summarize, the collection of seven TFBS groups may be functionally related to developmental processes. Specifically, when compared with a background gene list containing all the genes associated with the collection of seven TFBS groups, developmental process terms were enriched for the genes nearby the top accelerated binding sites among the seven groups (Fig. 9). Therefore, among the collection, the binding sites with strongest signals of accelerated evolution might be more crucial to the developmental processes. Together with the fact that a large proportion of HARs are neural enhancers and subject to accelerated evolution in humans and other primates[19–21], we conclude that regulatory sequences of neurodevelopmental genes may be the main target of accelerated evolution in primates.

Due to the scarcity of ChIP-seq data in non-human primates, we have used human-based TFBS annotations to infer accelerated evolution. It may limit our ability to detect accelerated evolution present in non-human primates but not in humans. Thus, our estimate of the number of accelerated TFBSs is likely to be conservative (Table 1). In future studies, it is of great interest to investigate accelerated evolution in TFBSs identified in non-human primates, highlighting the urgent need to perform high-throughput functional genomic experiments in our close relatives.

Compared to conserved genomic elements explored in previous studies of HARs, TFBSs may have a higher evolutionary turnover rate[30–33,38,39]. To alleviate the impact of evolutionary turnover on our analysis, we have only included primate genomes in the current study and filtered out low-quality alignments. Nevertheless, a small proportion of TFBSs identified in the human genome may still be subject to evolutionary turnover in other primates[39]. We expect that the evolutionary turnover of TFBSs in non-human primates may not lead to false positive results in our analysis. Indeed, conditional on the presence of a TFBS in the human genome, the evolutionary turnover of the TFBS in non-human primates is more likely to increase the substitution rate in the background lineage and hence makes our analysis conservative. Moreover, conditional on the loss of an old binding site in the human genome, the sequences would not be annotated as TFBSs in the human genome. Given that we used human genome annotation, those regions functional in background lineages but not in humans were not included in our analysis. Once ChIP-seq data become available in multiple non-human primates in the future, the conservativeness of our analysis may be alleviated by including only species where the TFBS of interest is detected.

Our pooling-based methods have a potential to be extended in future studies. For instance, if multiple TFBSs overlap with each other, our current methods cannot distinguish between TFBSs directly under accelerated evolution from those overlapping other accelerated TFBSs. To address this problem, we have used a heuristic method to remove overlapping TFBSs in the current study, which may reduce the number of TFBSs in our analysis. In the future, it is of great interest to develop a rigorous method for inferring accelerated evolution in overlapping TFBSs. Motivated by the recent success of evolution-based regression models[34,82–84], we propose that unifying our pooling-based methods and generalized linear models may be a promising direction to disentangle causal from correlational relationships in the analysis of accelerated evolution.

## Methods

### Genome alignment and TFBS annotation

We obtained the Multiz alignment of 46 vertebrate genomes from the UCSC Genome Browser[46]. Then, we extracted a subset of alignments for ten primate species from the 46-way Multiz alignment. The ten primate species and their genome assemblies included *Homo Sapiens* (hg19), *Pan troglodytes* (panTro2), *Gorilla gorilla* (gorGor1), *Pongo*

*abelii* (ponAbe2), *Macaca mulatta* (rheMac2), *Papio hamadryas* (papHam1), *Callithrix jacchus* (calJac1), *Tarsius syrichta* (tarSyr1), *Microcebus murinus* (micMur1), and *Otolemur garnettii* (otoGar1). Also, we downloaded 4,380,444 TFBSs for 161 transcription factors from the UCSC Genome Browser. These TFBSs were identified by ChIP-seq experiments in the ENCODE Project[45]. We extracted alignments of TFBSs across ten primate species using PHAST[85]. We removed the TFBSs overlapping with UTRs, CDSs, and previously identified HARs. To filter out low-quality alignments, we obtained informative alignment sites where unambiguous bases were found in at least five out of ten primate species in the Multiz alignment. We retained TFBSs with at least 50 informative alignment sites for downstream analysis.

### Previously defined HARs collection

We obtained a comprehensive list of previously defined HARs from https://docpollard.org/research/. We first combined the following genetic elements: Merged list of 2649 HARs(a set of HARs in non-coding regions built by Capra et al.[17]), 284 human accelerated elements in mammal conserved regions with adjusted *p*-value <0.05 (mapped to hg19 using the LiftOver tools on the UCSC genome browser), and 760 human accelerated elements in primate conserved regions with adjusted *p*-value <0.05 (mapped to hg19 using the LiftOver tools on the UCSC genome browser). Then we sorted and merged the bed file using bedtools/2.27.1.

### Group-level LRT for inferring accelerated evolution

We built a reference phylogenetic model using the alignment of ten primate genomes, assuming that the majority of TFBSs may not be subject to accelerated evolution. We first concatenated alignments of all TFBSs. We then fit a phylogenetic model to the concatenated alignment using the phangorn library in R[86]. In the phylogenetic model, we used the generalized time-reversible (GTR) substitution model to describe nucleotide sequence evolution and the discrete Gamma distribution with four rate categories to model substitution rate variation among nucleotide sites[42]. Also, we fixed the tree topology of the reference phylogenetic model to the one used from the UCSC Genome Browser (http://hgdownload.cse.ucsc.edu/goldenPath/hg19/multiz46way/46way.nh). We estimated model parameters, including branch lengths, the shape parameter of the discrete Gamma distribution, and parameters of the GTR substitution model, using the optim.pml function in phangorn.

Given the reference phylogenetic model, we used a customized R program based on phangorn to perform the group-level LRT. First, we concatenated alignments for each TFBS group separately. Then, we fit two group-level phylogenetic models to the concatenated alignment of each TFBS group. In the null model ($H_0$), we inferred a global scaling factor of branch lengths with maximum likelihood estimation and fixed all other model parameters to the ones in the reference phylogenetic model. We interpreted the estimated scaling factor as the relative substitution rate of TFBS sequences in both the foreground and the background lineage. In the alternative model ($H_a$), we estimated two scaling factors of branch lengths, $r_1$ and $r_2$, for the foreground and background lineages, respectively. The two scaling factors were interpreted as the relative substitution rates in the foreground and background lineages in the alternative model. For each TFBS group, we calculated a likelihood ratio statistic defined as the two-fold difference in the log likelihood between $H_a$ and $H_0$. Given the likelihood ratio statistic, we obtained a *p*-value for each TFBS group using a chi-square test with one degree of freedom. Finally, we calculated adjusted *p*-values using the Bonferroni correction.

From the group-level LRT, we found that TFBSs of 15 transcription factors showed elevated substitution rates in the human lineage. We further partitioned the TFBSs of the 15 transcription factors into 17 non-overlapping TFBS groups. These non-overlapping TFBS groups included genomic regions exclusively bound by one of the 15

transcription factors and two new TFBS groups: Pol III binding and POU5F1-NANOG binding. The Pol III binding group consisted of TFBSs bound by at least two of BDP1, BRF1, and POLR3G. Similarly, the group of POU5F1-NANOG binding consisted of TFBSs bound by both POU5F1 and NANOG. Then, we applied the group-level LRT again to the 17 non-overlapping TFBS groups and calculated adjusted *p*-values using the Bonferroni correction.

## Estimation of the number of TFBSs under accelerated evolution

We utilized the R program for the group-level LRT to perform the element-level LRT. To this end, we applied the phangorn package in R language[86,87] to the alignment of each individual TFBS separately, after filtering out TFBSs with less than 50 informative alignment sites. Then, we performed parametric bootstrapping at the group level to calculate a *p*-value for each TFBS. Specifically, we first fit the $H_0$ in the group-level LRT to the concatenated alignment of each TFBS group, which provided a global scaling factor to calibrate the branch lengths of the reference phylogenetic model. Second, we randomly sampled 10,000 TFBSs with replacement from each TFBS group and used the calibrated phylogenetic model to generate 10,000 simulated alignments of matched length. Third, we fit the element-level LRT to the simulated TFBS alignments from the same group, which provided an empirical null distribution of the likelihood ratio statistic for each TFBS group. Fourth, we compared the observed likelihood ratio statistic to the empirical null distribution to calculate a *p*-value for each TFBS. Finally, we fit a beta-uniform mixture model with probability density function (PDF)

$$f(x|a,\lambda) = \lambda + (1-\lambda)ax^{a-1} \qquad (1)$$

to *p*-values from each TFBS group[44]. We considered a statistic, $\hat{\pi}_{ub}$, from the beta-uniform mixture model as the upper bound of proportion or binding site without acceleration and, accordingly, $1 - \hat{\pi}_{ub}$ as the lower bound of proportion of accelerated TFBSs.

$$\hat{\pi}_{ub} = \hat{\lambda} + (1-\hat{\lambda})\hat{a} \qquad (2)$$

To build 95% confidence interval for $\hat{\pi}_{ub}$, we first searched for all values of $\lambda^{\star}$ and $a^{\star}$, such that

$$2(l(\hat{a},\hat{\lambda}|x) - l(a^{\star},\lambda^{\star}|x)) \le \chi^2_{2,1-\alpha} \qquad (3)$$

The 95% confidence interval for $\hat{\pi}_{ub}$ was calculated by combinations of $\lambda^{\star}$ and $a^{\star}$ which fell into the confidence interval.

$$\pi^{\star}_{ub} = \lambda^{\star} + (1-\lambda^{\star})a^{\star} \qquad (4)$$

## Simulations

We generated eight cases in which different lineages of primates were under accelerated evolution: (1) only human, (2) subtree of all the hominini(human, chimp), (3) subtree of human, chimp, gorilla, (4) subtree of all the apes(human, chimp, gorilla, orangutan), (5) only chimp, (6) only groilla, (7) only orangutan, (8) only macaque. For each case, folds of increase in substitution rates of accelerated lineage span from 1.2 to 5. We generated 10,000 ten-sequence alignments based on the reference model plus those assumptions. Each alignment is 200 bp long, which is the median length in TFBS data. We then compared the performance of our GroupAcc methods and traditional element-level LRT methods in detecting lineage-specific acceleration. Group-level LRT method and Phylogenetics-based mixture model were described in the former two sections. Traditional element-level LRT was implemented via the R program for the group-level LRT followed with Bonferroni correction to the *p*-values.

In scenario 1, we generated 10,000 200 bp alignments upon reference model and a scaled tree with increased branch length in lineages of each case. First, we applied the group-level LRT and phylogenetics-based mixture model to the simulated alignments, taking the accelerated lineage listed in each case (1–8) as fore-ground lineage, respectively. We compared the estimated fold of increase in substitution rate in foreground lineage with the scaling factor of the phylogenetic tree in simulation setting. We also compared the estimated number of elements under accelerated evolution from phylogenetics-based mixture model and element-level LRT methods. Second, the same methods were used with human as foreground lineage for all the cases. Cases 1–4 were designed to test the sensitivity of the methods to identify accelerated evolution when the foreground lineage (human) is truly under accelerated evolution. Cases 5–8 were designed to test the specificity of the methods when the foreground lineage (human) is mis-specified and not under accelerated evolution. We then compared the estimated fold of increase in substitution rate in foreground lineage with the scaling factor of phylogenetic tree in simulation setting. We also compared the estimated number of elements under accelerated evolution from phylogenetics-based mixture model and element-level LRT methods.

The second scenario considered heterogeneity of evolutionary dynamics in each binding site: only parts of each binding site (L: portion of each binding site under accelerated evolution) were under accelerated evolution. We simulated 10,000 ten-sequence alignments representing 10,000 binding sites in one group, each binding site is 200 bp long ($200 \times L$ bp generated from a scaled tree with X-fold increase in branch length of the lineage shown in the cases, $200 - 200 \times L$ bp generated from unscaled tree). We analyzed the data with our mixture model to see if our method could estimate the proportion of binding sites with accelerated evolution accurately. In addition, we tested with group-level LRT method to see if our methods could detect group-level signals and estimate the fold of increase in substitution rates when the acceleration only happens in specific positions or motifs.

The third scenario considered heterogeneity in groups of binding sites: only certain numbers of binding sites (M: proportion of binding sites in a group under accelerated evolution) in one group have accelerated evolution in a specific lineage, while the other binding sites do not have accelerated evolution. We simulated 10,000 elements in a group, $10,000 \times M$ elements from a scaled tree, while $10,000 - 10,000 \times M$ from unscaled tree. Each element is 200 bp long. We analyzed the data with our mixture model to see if our method can estimate the number of binding sites with accelerated evolution accurately. In addition, we tested with group-level LRT method to see if our methods can detect group-level signals and estimate the fold of increase in substitution rates when the acceleration only happens in parts of the binding sites in a group.

## Reduction of redundancy in the 15 TFBS groups

From Group-level LRT, we found 15 groups of TFBSs with accelerated evolution in human. However, there is redundancy among the data possibly because the transcription factors share a considerable proportion of binding sites.

Some of the groups have similar biological functions, for example, BDP1, BRF1 and POLR3G are key factors in the Pol III transcription machinery; POU5F1 and NANOG are necessary regulators in ES cell pluripotency and self-renewal. To identify the evolutionary forces in the colocalization of transcription factors, we defined two new TFBS groups. The Pol III binding sites, were defined as the binding sites occupied by at least two out of the three transcription factors related to Pol III (BDP1, BRF1 and POLR3G). To define the POU5F1-NANOG binding, we obtained the intersecting regions of POU5F1 and NANOG binding sites.

To remove redundancy in overlapping binding sites, we then got the non-overlapping regions bound by merely BDP1, BRF1 or POLR3G. For each of the other 12 TFBS groups with accelerated evolution in human, we obtained the entries that don't overlap with any of BDP1, BRF1, POLR3G or the rest 11 TFBS groups. Then we ran the group-level LRT again for the 15 non-overlapping TFBS groups and 2 newly-defined TFBS groups.

### Inference of lineages with accelerated evolution

We utilized the alternative model ($H_a$) in the group-level LRT to search for lineages associated with accelerated evolution. To this end, we fit the group-level $H_a$ with seven different foreground lineages to the concatenated alignment of each TFBS group (Fig. 7). The seven foreground lineages corresponded to all monophyletic clades that included humans. For each TFBS group and foreground lineage, we used the BIC as a measure of goodness-of-fit,

$$BIC = -2l + k \log(n),  \qquad (5)$$

where $l$ is the log likelihood of the group-level $H_a$, $k$ is the number of model parameters, and $n$ is the sample size. Because the group-level $H_a$ included two parameters ($r_1$ and $r_2$), we set $k$ to 2. Also, we assumed that $n$ could be approximated by the total number of bases in the concatenated alignment of each TFBS group. For each TFBS group, we considered the foreground lineage with the highest BIC to be the best-fit lineage.

### Detection of selection pressure and GC-biased gene conversion

To investigate if accelerated evolution in TFBSs was driven by positive selection, we used the INSIGHT model to infer positive selection on the seven accelerated, non-overlapping TFBS groups in the human lineage[50–52]. We obtained INSIGHT2, a highly efficient implementation of the INSIGHT model, from https://github.com/CshlSiepelLab/FitCons2. Then, we applied INSIGHT2 to each TFBS group under accelerated evolution and the collection of all TFBSs from ENCODE. INSIGHT2 provided $D_p$ and $SE[D_p]$, that is, the expected number of adaptive substitutions per kilobase and its standard error, as well as $\rho$ and $SE[\rho]$ which quantified the fraction of sites under selection within functional elements and its standard error. We performed the Wald test to examine if $D_p$ was significantly different from 0 for each TFBS group. Under the null hypothesis of $D_p = 0$, we assumed that the z-statistic, $\frac{D_p}{SE[D_p]}$, asymptotically followed a 50:50 mixture of a point probability mass at 0 and a half standard normal distribution[88]. We conducted comparisons of $\rho$ among the seven accelerated TFBS groups and the collection of all TFBSs from ENCODE (Supplementary Table 2). To identify the role of GC-biased gene conversion in the accelerated evolution of the seven TFBS groups, we used the phastBias model to infer gene conversion disparity $B$ in the lineage where accelerated evolution occurred, identical to the best-fit lineage found in model comparison (Fig. 7).

### Functional enrichment analysis of accelerated TFBS associated genes

To investigate specific functions of the accelerated binding sites in each group, we performed functional enrichment analysis of the accelerated TFBS-associated genes. We first extracted the TFBSs with significant results in the phylogenetics-based mixture model and referred them to the top accelerated TFBSs. Then we identified the potential target genes of the seven TFBS groups as well as top accelerated bindings sites among the seven groups using GREAT with default settings. For each of the seven groups, we performed GO enrichment analysis using clusterProfiler on the genes associated with top accelerated binding sites, with the genes associated with the TFBS group, respectively, as background. Besides, we performed GO enrichment analysis on the genes associated with the concatenation of all the top accelerated binding sites across seven groups, with genes associated with all TFBSs as background, With the clusterProfiler package, the significance of enrichment test for GO terms under biology process subontology was estimated by hypergeometric distribution and then adjusted by Bonferroni correction. The redundant GO terms were trimmed by applying the *simplify* function to remove terms among which semantic similarities were higher than 0.7. Significant terms after Bonferroni correction were shown in the Fig. 9, while the complete list of significant GO biological process terms with corrected *p*-value <0.05 are available in the Supplementary Data 2. Benjamini–Hochberg correction was also used in to the GO results and the significant results are listed in Supplementary Data 2.

### Primate ChIP-seq data

We obtained a list of histone H3 lysine 27 acetylation (H3K27ac) enriched regions in human, chimpanzee and rhesus macaque brain[59] from NCBI GEO Series GSE67978. The H3K27ac enriched regions were predicted to be active cis-regulatory elements (CREs). Since hg19 has been the reference genome in Multiz alignment, the annotated regions were first mapped to hg19 using the LiftOver tools on the UCSC genome browser and then processed using bedtools/2.27.1. We extracted the alignments of those annotated regions from the Multiz alignment. We applied the group-level LRT method to the CREs in human brain with human as the foreground lineage, to the CREs in chimpanzee brain with chimpanzee as the foreground lineage and to the CREs in rhesus macaque brain with rhesus macaque as the foreground lineage.

We also obtained a list of trimethylated lysine 4 of histone H3 (H3K4me3) enriched regions and H3K27ac enriched regions in liver of 20 mammals including human and rhesus macaque[60] (Accession E-MTAB-2633). The annotation of regions was first mapped to hg19 using the LiftOver tools on the UCSC genome browser. Then we sorted and merged the bed file using bedtools/2.27.1. The regions were classified into active gene promoters and enhancers. Enhancers were identified by regions only enriched for H3K27ac, while promoters defined as regions containing both H3K27ac and H3K4me3. We obtained the bed files of promoters and enhancers using intersect function in bedtools/2.27.1.

We obtained a list of CTCF tissue-specific binding sites[61] and downloaded the annotated files from ENCODE.

### Reporting summary

Further information on research design is available in the Nature Portfolio Reporting Summary linked to this article.

## Data availability

Human TFBS annotation were downloaded from Txn Factor ChIP Track on UCSC genome browser. Primate alignments were extracted from Multiz alignment of 46 vertebrate genomes from the UCSC Genome Browser (http://hgdownload.cse.ucsc.edu/goldenPath/hg19/multiz46way/). Reference model built with concatenated alignments of all TFBSs was uploaded to https://github.com/May-BG/GroupAcc[89]. TFBS groups with accelerated evolution in primates were uploaded to the Github page. Previously defined HARs collection were downloaded from https://docpollard.org/research/. ChIP-seq data of primates' brains were downloaded from NCBI GEO Series GSE67978 and Accession E-MTAB-2633. Human tissue-specific CTCF binding sites information were downloaded from ENCODE with the accession numbers from Supplementary Data 3.

## Code availability

GroupAcc and companion data are available at https://github.com/May-BG/GroupAcc[89].

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

## Acknowledgements

The authors thank Adam Siepel, Ilan Gronau, Zhihan Liu and Ritika Ramani for useful discussions. Research reported in this publication was supported by the National Institute of General Medical Sciences of the National Institutes of Health under Award Number R35GM142560 (to Y.H.), the Pennsylvania State University (to X.Z. and Y.H.) and a postdoctoral fellowship from the Harvard University (to B.F.). The content is solely the responsibility of the authors and does not necessarily represent the official views of the National Institutes of Health.

## Author contributions

Y.H. conceived of and supervised the project. X.Z. conducted all analyses with contributions from Y.H. and B.F. X.Z. and Y.H. wrote the manuscript. B.F. contributed to simulation analyses and visualization. All authors provided comments and revisions on drafts and approved the final paper.

## Competing interests

The authors declare no competing interests.
