## [Peer Review File · Nature Communications]

Transcription factor binding sites are frequently under accelerated evolution in primatesReviewers' Comments:

Reviewer #1:

Remarks to the Author:

In this work, Zhang & Huang search for transcription factors (TFs) whose binding sites are undergoing accelerated evolution in humans or primates. There has been impactful past work on identifying human accelerated regions (HARs) in the genome, with the hope of finding genomic segments that especially shaped human evolution. Here, the authors sought to find out if there are TFs whose binding sites may similarly be under accelerated evolution and hence shed insight on unique aspects of human evolution. They developed two statistical methods to detect accelerated evolution in TF binding sites as a group. Based on use of these methods, the authors claim that collectively the TFBS under accelerated evolution are more abundant than the "HARs" previously reported in the literature. An interesting finding made by the authors is that Pol III binding sites are under lineage-specific positive selection.

An important aspect of the method used in this work is that it detects accelerated evolution in genomic elements that may be under, by relaxed purifying selection or weak positive selection. Basically, it is a harder task in some ways to find accelerated TFBS (compared to finding conserved regions that are human-accelerated) because they are less well conserved and likely under weaker acceleration relative to conserved elements; the authors address this methodological challenge.

Major comments:

The two methods need extensive statistical assessment on synthetic data based on realistic models of TFBS evolution, under varying assumptions and parameters of the lineage-specific evolutionary dynamics. Without such extensive evaluation (and comparison to suitable baselines) the contribution of the two new methods is unclear.

It is not clear if the effects of TFBS turnover have been addressed appropriately. Studies of enhancer evolution in Drosophilids have repeatedly found that individual binding sites (defined as motif matches) undergo substantial turnover while the overall binding site composition and function of the enhancers (~500-1000 bp) are conserved. This is a crucial point to address because turnover may artificially elevate the inferred evolutionary rate in an enhancer and the TFBS analyzed by the authors are of length-scale similar to enhancers.

Both models assume that TFBS evolution can be well modeled by assuming a uniform scaling of substitution rate on the foreground branches. This sort of modeling, while meritorious in its simplicity, ignores the heterogeneity of evolutionary constraints within the TFBS (ChIP peak), and even within a single motif match of a TFBS. There is a substantial literature that has looked into the position-specificity of evolutionary dynamics in binding sites (motif matches) and the fact that selection acts on the entire TFBS rather than on individual positions within it. Please comment on the implications of the simplification.

The removal of binding sites overlapping more than one group (e.g., ChIP peaks of two different TFs that overlap each other) may introduce an ascertainment bias of unknown degree, especially if the overlap is lineage-specific (does this happen?). Please comment on this.

Am I correct in reading Figure 2 and associated text as meaning that groups of TFBS with relative substitution rate barely above 1 (4-5 of these seven TFs) may also be considered as undergoing accelerated evolution? This casts serious doubts into the interpretation of the "downstream analysis", vis-à-vis the meaning of "accelerated evolution".

The claim associated with Figure 3 appears uninteresting. After having claimed that they identified a

small set of TFs with human-specific accelerated evolution (previous subsection), this analysis takes a step back and says that those findings may not be human-specific after all, and may be an acceleration in a broader part of the tree. What is ruled out? I suppose with some creativity in defining groups of genomic segments, any part of the tree may be found to have certain groups under accelerated evolution. Can the authors clarify why their finding is interesting?

Regarding the analysis shown in Table 1, could the authors provide estimates of uncertainty in the proportions reported? Since this is based on a beta-uniform mixture fit to a collection of p-values, with the beta itself having free parameters, it is entirely possible that there is substantial uncertainty in the proportions.

Comparison of the number of TFBS under accelerated evolution against the number of HARs is fraught with potential misinterpretation. The authors have not done enough to convince the reader that those two numbers, with their own procedures of derivation and being counts of rather different entities, are indeed comparable.

The GO enrichment analysis of TFBS groups with supposed accelerated evolution is interesting. What happened to the BDP1 enrichment results?

Minor comments:

The presentation should clarify early on that the term TFBS is used to mean ChIP peaks. Technically, a ChIP peak is an approximate region for a TF binding, where multiple binding sites of the same TF and often sites for multiple TFs may be present. In reading the first section of results, I was under the impression that the TFBS being analyzed means motif matches within ChIP peaks, not the entire peak, which has heterogeneous evolutionary dynamics at different positions. The usage adopted by the authors is not unusual, and they do not need to change it, just clarify in the beginning.

Reviewer #2:

Remarks to the Author:

In this manuscript, Zhang and Huang study transcription factor binding regions which have evolved rapidly in the lineage leading to humans. This work builds on a tradition of identifying and characterising human accelerated regions. Unlike many previous papers which focused on sequences defined by evolutionary conservation (outside humans), the authors here focus on sequences defined by function. This is an interesting approach, and the work is overall interesting.

This work is related to recent work of ours, so I will sign my review: Marc Robinson-Rechavi. In a general way, I would like to point out that in Science Advances 6: eabc9863, we reported that "the TFBSs of cell types detected under selection do not necessarily evolve faster", which seems relevant to the discussion of results here.

Major comments:

The study only considers cases of human lineage acceleration. To conclude about human evolution, it would be very helpful to contrast results to those centering on other lineages, be they other primates or mouse, for which similarly abundant ChIP-seq data is available.

Acceleration might be due to positive selection, relaxed purifying selection, or biased gene conversion, as shown in many of the papers cited. This possibility has to be considered, both in the analysis and in the discussion.

Differences in detection of selection between site categories could be due to the number of nucleotides

or sites concerned, i.e. sample size. The authors should check for this, for example by down-sampling Pol III sites and verifying if selection is still detected.

I did not understand what the authors are testing with the GO and Reactome enrichments. By testing the genes close to specific TFs, they recover the function of these TFs biased by the tissues or cell types in which the ChIP-seq was done, but we do not learn anything specific about the accelerated or selected TFBSs. I suggest to instead perform enrichment test with background all genes with a given TF, foreground all genes with that TF accelerated.

In the Science Advances paper cited above, we compared selection of TFBSs according to the organ or tissue where they were active. It would be very interesting to see the same here, and to compare results. This would also allow to meaningfully test subsets of TFBSs for positive selection, and maybe characterize the genes which are involved in tissue-specific adaptive acceleration.

I don't understand this statement: "To the best of our knowledge, our methods are the first statistical framework dedicated to infer weakly accelerated evolution." Maybe it is correct in the very specific context of non coding sequences in human evolution? Please clarify.

Minor comments:

Line 109, "Unlikely" should be "Unlike"

All statements of the form $x\% y$ should be of the form $x\% \text{ of } y$, e.g. "78% of Pol III binding sites".

Line 174 "more" is repeated.

Bonferroni takes an uppercase B.

We thank the two reviewers for reading our manuscript and offering thoughtful suggestions to improve the manuscript. We have made additional analyses, edited the manuscript, and provided a point-to-point response based on the reviewers' comments.

REVIEWER COMMENTS

Reviewer #1 (Remarks to the Author):

In this work, Zhang & Huang search for transcription factors (TFs) whose binding sites are undergoing accelerated evolution in humans or primates. There has been impactful past work on identifying human accelerated regions (HARs) in the genome, with the hope of finding genomic segments that especially shaped human evolution. Here, the authors sought to find out if there are TFs whose binding sites may similarly be under accelerated evolution and hence shed insight on unique aspects of human evolution. They developed two statistical methods to detect accelerated evolution in TF binding sites as a group. Based on use of these methods, the authors claim that collectively the TFBS under accelerated evolution are more abundant than the "HARs" previously reported in the literature. An interesting finding made by the authors is that Pol III binding sites are under lineage-specific positive selection.

An important aspect of the method used in this work is that it detects accelerated evolution in genomic elements that may be under, by relaxed purifying selection or weak positive selection. Basically, it is a harder task in some ways to find accelerated TFBS (compared to finding conserved regions that are human-accelerated) because they are less well conserved and likely under weaker acceleration relative to conserved elements; the authors address this methodological challenge.

We thank the reviewer for your kind comments and constructive feedback.

Major comments:

1 The two methods need extensive statistical assessment on synthetic data based on realistic models of TFBS evolution, under varying assumptions and parameters of the lineage-specific evolutionary dynamics. Without such extensive evaluation (and comparison to suitable baselines) the contribution of the two new methods is unclear.

Following the suggestions, we simulated synthetic data based on the reference phylogenetic model of TFBS evolution, plus various scenarios of lineage-specific evolutionary dynamics. We then assessed the performance of the GroupAcc methods and traditional single-level method in recovering the lineage-specific evolutionary patterns, especially the weakly accelerated evolution. We have added the simulation results in L. 105-164 and methods in L. 468-512.

To consider realistic models, we assumed that the majority of TFBSs were not under accelerated evolution in human. Thus, leveraging the concatenated alignments of all TFBSs, we fit a reference phylogenetic model that represented the overall pattern of TFBS evolution without

accelerated evolution. The parameters estimated in the reference phylogenetic model included the branch lengths of a phylogenetic tree, the gamma shape parameter for rate variation among nucleotides, and the parameters of the general time reversible substitution model.

In the simulations, we proposed three scenarios of lineage-specific evolutionary dynamics. In the first scenario, all the binding sites in one group are under accelerated evolution in a specific lineage. The second scenario considered position-specificity or heterogeneity of evolutionary patterns in each binding site: only parts of each binding site (for example, motif) have accelerated evolution. The third scenario considered heterogeneity in groups of binding sites: only certain numbers of binding sites in one group have accelerated evolution in a specific lineage, while the other binding sites do not have accelerated evolution. Under each scenario, we verified the ability of the Group-level LRT method to detect accelerated evolution in a specific lineage and estimate the substitution rate increase (r_1/r_2). Consequently, we summarized that the phylogenetics-based mixture model outperformed traditional single-level LRT in estimating the number of elements under accelerated evolution in each lineage.

Under the first scenario, all the 200-bp binding sites in one group were assumed to be under accelerated evolution in a defined lineage as each case showed. We generated 10,000 20-bp alignments upon the reference model and a scaled tree with increased branch length in lineages of each case (cases 1-8). Then group-level LRT and phylogenetics-based mixture model were applied to the simulated alignments in ease case.

With foreground lineage matching with the accelerated lineage in each case, the group-level LRT method was able to tell the presence of accelerated evolution at the group level and accurately estimate the fold of increase in substitution rate in foreground lineages (r_1 / r_2), even given weak accelerated evolution when the fold of increase in substitution rates is slightly larger than 1 (Fig. 2A). The GroupAcc model performed better than element-level LRT in estimating the number of elements under accelerated evolution. (Fig. 2B).

We also tested if the model could detect accelerated evolution in a tip if a subtree containing the tip is under accelerated evolution (Fig. 3). In cases (1), (2), (3) and (4), when accelerated evolution happened in lineages such as human or subtrees containing human, taking human as foreground lineage, the GroupAcc methods were able to identify the presence of accelerated evolution in human and estimate the number of elements under accelerated evolution in human with higher accuracy compared to traditional element-level LRT method (Fig. 7). In cases (5), (6), (7) and (8), when accelerated evolution occurred in lineages other than human, the GroupAcc methods were able to identify the fact that human is not undergoing accelerated evolution (Fig. 3).

Figure 2: Simulation results of scenario 1 with foreground lineage matching the accelerated lineage in each case. (A) X-axis shows the scaling factor of foreground lineage branch length in simulation setting, which is the real fold of increase in foreground lineage. Y-axis shows the fold of increase in foreground lineage estimated from group-level LRT. (B) Comparison of accuracy estimating the number of elements under accelerated evolution between GroupAcc and element-level LRT method. Blue curves are the accuracy of GroupAcc. Red curves are the accuracy of element-level LRT.

Figure 3: Simulation results of scenario 1 using human as foreground lineage. (A) X-axis shows the scaling factor of accelerated lineage branch length in simulation setting, which is the real fold of increase in accelerated lineage. Y-axis shows the fold of increase in human estimated from group-level LRT. (B) Comparison of the estimated number of elements under accelerated evolution in human between GroupAcc and element-level LRT method. Blue curves are the estimates of GroupAcc. Red curves are the estimates of element-level LRT.

We validated the ability of our methods to identify lineage-specific acceleration when only part of the TFBS is under accelerated evolution from simulation scenario 2. We generated 10,000 200-bp alignments standing for elements. Each alignment was composed of $L * 200$ bp generated with a scaled tree (with substitution rate increase) and $200 - L * 200$ bp generated from an unscaled tree (without substitution rate increase). Given that $L = 0.1, 0.2, 0.5, 0.8$, the group-level LRT was able to identify the presence of accelerated evolution, even under weak acceleration when the fold of substitution rate increase in foreground lineage was only 1.2 (Fig. 4A). The GroupAcc method outperformed the element-level LRT method in estimating the number of elements under accelerated evolution (Fig. 4B).

Figure 4: Simulation results of scenario 2. (A) Accuracy of GroupAcc in estimating the fold of increase in foreground lineage given different portions of each binding site under accelerated evolution. X-axis shows the scaling factor of accelerated lineage branch length in simulation setting, which is the real fold of increase in accelerated lineage. Y-axis shows the accuracy of estimating the fold of increase. The weighted estimates the fold of increase in foreground lineage across the whole group of binding sites is calculated by $(L * \hat{r}_1 / \hat{r}_2 + 1 - L)$. The accuracy of estimating the fold of increase is calculated as $\frac{\text{The weighted estimates the fold of increase in foreground lineage}}{r_1 / r_2 \text{ in the simulation setting}}$. (B) Comparison of performance estimating the number of elements under accelerated evolution between GroupAcc and element-level LRT method. Blue curves are the estimated numbers of GroupAcc. Red curves are the estimated numbers of element-level LRT.

Under the third scenario, a specific proportion of binding sites ($M = 0.1, 0.2, 0.5, 0.8$) in a group were under accelerated evolution. This scenario considered the heterogeneity of evolutionary dynamics in multiple binding sites of one transcription factor. We found group-level LRT method was able to tell the presence of accelerated evolution at the group level and estimate the fold of increase in substitution rate of foreground lineages, even when the fold of increase in substitution rates of foreground lineage was slightly larger than 1 (Fig. 5). The GroupAcc model performed better than element-level LRT in estimating the number of elements under accelerated evolution (Fig. 5).

Figure 5: Simulation results of scenario 3. (A) Accuracy of GroupAcc in estimating the fold of increase in foreground lineage given different portions of elements in a group under accelerated evolution. X-axis shows the scaling factor of accelerated lineage branch length in simulation setting, which is the real fold of increase in accelerated lineage. Y-axis shows the accuracy of estimating the fold of increase. The weighted estimates the fold of increase in foreground lineage across the whole group of binding sites is calculated by $(M * \hat{r}_1 / \hat{r}_2 + 1 - M)$. The accuracy of estimating the fold of increase is calculated as $\frac{\text{The weighted estimates the fold of increase in foreground lineage}}{r_1 / r_2 \text{ in the simulation setting}}$. (B) Comparison of performance estimating the number of elements under accelerated evolution between GroupAcc and element-level LRT method. Blue curves are the estimated numbers of GroupAcc. Red curves are the estimated numbers of element-level LRT.

2 It is not clear if the effects of TFBS turnover have been addressed appropriately. Studies of enhancer evolution in Drosophilids have repeatedly found that individual binding sites (defined as motif matches) undergo substantial turnover while the overall binding site composition and function of the enhancers (~500-1000 bp) are conserved. This is a crucial point to address because turnover may artificially elevate the inferred evolutionary rate in an enhancer and the TFBS analyzed by the authors are of length-scale similar to enhancers.

We acknowledge that TFBSs may have a higher evolutionary turnover rate as compared to conserved genomic elements explored in previous studies of HARs. We took some measures to mitigate the effects of TFBS turnover: 1). We included only primate genomes in the study, which will include fewer turnover events. 2). We filtered out low-quality alignments by requiring each TFBS to at least 50 informative sites where unambiguous bases were found in at least five out of ten primate species in the alignment. Nevertheless, a small proportion of TFBSs identified in the human genome may still be subject to evolutionary turnover in other primates.

For the following reasons, we expect our analysis to be robust and conservative and TFBS turnover will not lead to false positive results in the study. First, conditional on the presence of a TFBS in the human genome, the evolutionary turnover of the TFBS in non-human primates is more likely to increase the substitution rate in the background lineage and hence makes our analysis conservative. Moreover, conditional on the loss of an old binding site in the human genome, the sequences would not be annotated as TFBS in the human genome. Given that we used human genome annotation, those regions functional in background lineages but not in humans were not included in our analysis. In the revision, we have clarified the above comments in L. 365-368 and L. 373-376.

3 Both models assume that TFBS evolution can be well modeled by assuming a uniform scaling of substitution rate on the foreground branches. This sort of modeling, while meritorious in its simplicity, ignores the heterogeneity of evolutionary constraints within the TFBS (ChIP peak), and even within a single motif match of a TFBS. There is a substantial literature that has looked into the position-specificity of evolutionary dynamics in binding sites (motif matches) and the fact that selection acts on the entire TFBS rather than on individual positions within it. Please comment on the implications of the simplification.

By assuming a uniform scaling of substitution rate in the sequence of the binding site without assigning another scale to the specific positions, we might dilute the signals of accelerated evolution. Therefore, our findings are more conservative. In addition, in simulation scenario 2 in the revised manuscript, we have validated the ability of our methods to identify lineage-specific acceleration when only part of TFBS is under accelerated evolution. Specifically, we have generated 10,000 200-bp elements. Each of them has $L \times 200$ bp generated with a scaled tree (with substitution rate increase) and $200 - L \times 200$ bp generated from an unscaled tree (without substitution rate increase). Consequently, when $L=0.1, 0.2, 0.5$ or 0.8 , the group-level likelihood ratio test is able to identify the presence of accelerated evolution, even in weak acceleration scenarios when the fold of substitution rate is only 1.2. In the revision, we have discussed the

implications of the simplification in L.153-156. Simulation of scenario 2 was described in L.145-156 and L. 494-503.

4 The removal of binding sites overlapping more than one group (e.g., ChIP peaks of two different TFs that overlap each other) may introduce an ascertainment bias of unknown degree, especially if the overlap is lineage-specific (does this happen?). Please comment on this.

There could be overlapping binding sites shared by more than one transcription factors. Diehl and Alan (2018, *Nucleic acids research*, 46:1878-1894) revealed species-specific transcription factor co-binding patterns in human and mouse. Since we used primate phylogeny, species-specific transcription factor co-binding events would be fewer than those between human and mouse.

For the 15 TFBS groups showing substitution rate increase in human, we did a literature search and checked their overlap patterns. We kept the overlapping regions bound by transcription factors sharing similar biological functions or forming complex together. Because BDP1, BRF1 and POLR3G are components of Pol III binding sites, we defined a new group, Pol III binding sites, composed of genomic regions bound by at least two of the three transcription factors. Because POU5F1 and NANOG form a protein complex together, we defined another group, POU5F1-NANOG binding sites composed of the regions bound by both transcription factors.

From a collection of the 15 groups and two newly defined groups, we removed all the binding sites overlapping more than one TFBS groups. Group-level LRT were then applied to the 17 non-overlapping groups.

The method we use in this study is unable to distinguish between TFBSs directly under accelerated evolution from those overlapping other accelerated TFBSs. However, our heuristic method which removes overlapping TFBSs may reduce the number of TFBSs and provide a conservative estimate of the number of TFBSs under accelerated evolution. We planned to infer accelerated evolution in overlapping TFBSs by unifying GroupAcc and generalized linear models,

In the manuscript, we have discussed the implications of the simplification in the L.380-389.

5 Am I correct in reading Figure 2 and associated text as meaning that groups of TFBS with relative substitution rate barely above 1 (4-5 of these seven TFs) may also be considered as undergoing accelerated evolution? This casts serious doubts into the interpretation of the “downstream analysis”, vis-à-vis the meaning of “accelerated evolution”.

Yes, the reviewer is correct. The binding sites of all seven TFs are considered as undergoing accelerated evolution and are used for downstream analysis. Some of them have weaker acceleration in the human lineage, but their folds of increase in substitution rates in human

lineage are larger than 1.05, still significantly different from 1 after Bonferroni correction based on the group-level likelihood ratio test (now in Supplementary Table 1). The ability to detect weaker acceleration is one of the advantages of our methods. In the revision, we have added the statement “weakly accelerated evolution” before the “downstream analysis” (L.194).

Supplementary Table 1: Non-overlapping TFBS groups under accelerated evolution. r_1 and r_2 are the relative substitution rates of a TFBS group in the human lineage and in other primates. P -values are calculated from likelihood ratio test. The ratio (r_1/r_2) indicates the fold of increase in substitution rate in the human lineage.

Genomic elements	r_1	r_2	P -value	r_1/r_2
Pol III binding	2.30	1.41	0	1.63
BDP1	1.34	1.12	1.41E-06	1.19
POU5F1-NANOG binding	0.91	0.83	0.002	1.10
POU5F1	1.03	0.94	1.5E-05	1.09
FOXP2	0.99	0.92	0	1.08
NANOG	0.90	0.85	0.004	1.05
NRF1	1.16	1.11	0.03	1.05

6 The claim associated with Figure 3 appears uninteresting. After having claimed that they identified a small set of TFs with human-specific accelerated evolution (previous subsection), this analysis takes a step back and says that those findings may not be human-specific after all, and may be an acceleration in a broader part of the tree. What is ruled out? I suppose with some creativity in defining groups of genomic segments, any part of the tree may be found to have certain groups under accelerated evolution. Can the authors clarify why their finding is interesting?

In the revised manuscript, we have added simulations of scenario 1 (also shown in new Figure 3A) to test the model performance using human as foreground lineage when different lineages are under accelerated evolution (case 1-8). Group-level LRT with human as foreground lineage would identify that there is substitution rate increase in human when accelerated evolution occurred in human (case 1) or any monophyletic clades containing human (case 2-4), but not when accelerated evolution took place in merely non-human primates (case 5-8).

As for the empirical analysis, we started by identifying the substitution rate increase in human by taking human as foreground lineage. From 161 groups of TFBSs, we found seven non-overlapping groups of TFBSs with the substitution rate increase in human. By doing that, TFBSs that were under accelerated evolution specifically in non-human primates were ruled out.

However, we were not sure whether they are under human-specific accelerated evolution or under accelerated evolution in both human and other primates. Previous studies found that many HARs may also undergo accelerated evolution in multiple lineages of apes (Lindblad-Toh,

Garber et al. 2011, Kostka et al. 2018). Simulation results also suggested the possibility that some TFBSs might be under accelerated evolution in the monophyletic clades that included the human lineage.

To characterize when the acceleration occurred, we applied the group-level LRT with different foreground lineages to each of the seven TFBS groups and compared their goodness-of-fit. We were not ruling out anything; instead, we were trying to find the exact clade when the accelerated evolution took place. We found the accelerated evolution of TFBSs occurred in hominini, apes, and old-world monkey.

The core interesting finding in this study is TFBSs are frequently under accelerated evolution in primates, not only in humans, as the title states. To make the main text logically sound, we have also added explanations in L.197-200 in the revision.

7 Regarding the analysis shown in Table 1, could the authors provide estimates of uncertainty in the proportions reported? Since this is based on a beta-uniform mixture fit to a collection of p-values, with the beta itself having free parameters, it is entirely possible that there is substantial uncertainty in the proportions.

In the revision, we have provided the estimates of uncertainty of the proportions in L. 448-467 and reported the results in the supplementary Table 4. To do so, we have adopted *Bum* function in the *ClassComparison* package and *optim* function in R to minimize the negative log-likelihood of the model with parameters.

We fit a beta-uniform mixture model (Pounds and Morris 2003) to the empirical p-values for each binding site in a group with pdf

$$f(x|a, \lambda) = \lambda + (1 - \lambda)ax^{a-1}$$

We considered a statistic, $\hat{\pi}_{ub}$ from the beta-uniform mixture model as the upper bound of proportion or binding site without acceleration and, accordingly, $1 - \hat{\pi}_{ub}$ as the lower bound of proportion of accelerated TFBSs.

$$\hat{\pi}_{ub} = \hat{\lambda} + (1 - \hat{\lambda})\hat{a}$$

To build 95% confidence interval for $\hat{\pi}_{ub}$, we first searched for all values of λ^* and a^* such that

$$2 \left(l(\hat{a}, \hat{\lambda} | x) - l(a^*, \lambda^* | x) \right) \leq \chi_{2, 1-\alpha}^2$$

The 95% confidence interval for $\hat{\pi}_{ub}$ was calculated by combinations of λ^* and a^* which fell into the confidence interval.

The proportion of accelerated elements in a group is already a lower bound of elements generated from the alternative hypothesis.

$$\pi_{ub}^* = \lambda^* + (1 - \lambda^*)a^*$$

Supplementary Table 4: Estimates and confidence Intervals from the mixture model.

Genomic elements	$1 - \hat{\pi}_{ub}$	95%CI of $1 - \pi_{ub}$	\hat{a}	95%CI of a	$\hat{\lambda}$	95%CI of λ
PoI III binding	0.78	(0.74, 0.81)	0.22	(0.19, 0.26)	8.0E-6	(3.06E-7, 3.42E-5)
BDP1	0.20	(0.18, 0.24)	0.30	(0.22, 0.37)	0.71	(0.71, 0.72)
POU5F1-NANOG binding	0.08	(0.04, 0.11)	0.43	(0.30, 0.66)	0.86	(0.65, 0.94)
POU5F1	0.10	(0.07, 0.15)	0.38	(0.35, 0.47)	0.83	(0.71, 0.89)
FOXP2	0.27	(0.25, 0.29)	0.52	(0.51, 0.54)	0.43	(0.37, 0.49)
NANOG	0.26	(0.23, 0.29)	0.55	(0.47, 0.55)	0.42	(0.37, 0.56)
NRF1	0.25	(0.22, 0.29)	0.45	(0.41, 0.52)	0.54	(0.38, 0.54)

8 Comparison of the number of TFBS under accelerated evolution against the number of HARs is fraught with potential misinterpretation. The authors have not done enough to convince the reader that those two numbers, with their own procedures of derivation and being counts of rather different entities, are indeed comparable.

We thank the reviewer for this comment. We have estimated the lengths of accelerated TFBSs. The total length of accelerated TFBSs identified via our methods is 1,573kb. The length of HARs identified in previous methods is around 720kb. In the revised manuscript, we have added this comparison in L. 237-238.

We acknowledge that accelerated TFBSs and HARs have some differences. HARs are regions under strongly accelerated evolution found via traditional element-level LRT, while accelerated TFBSs are under weakly accelerated evolution identified via group-level methods. We are using the abundance of HARs as the reference to emphasize the abundance of elements under weakly accelerated regions

The key information we want to convey is that there could be more abundant regions under weakly accelerated evolution relative to the regions under strong acceleration.

9 The GO enrichment analysis of TFBS groups with supposed accelerate evolution is interesting. What happened to the BDP1 enrichment results?

We did not see GO-term significantly enriched with BDP1-associated genes, possibly because BDP1 bind fewer elements and the sample size is too small for enrichment.

Minor comments:

The presentation should clarify early on that the term TFBS is used to mean ChIP peaks. Technically, a ChIP peak is an approximate region for a TF binding, where multiple binding sites of the same TF and often sites for multiple TFs may be present. In reading the first section of results, I was under the impression that the TFBS being analyzed means motif matches within ChIP peaks, not the entire peak, which has heterogeneous evolutionary dynamics at different positions. The usage adopted by the authors is not unusual, and they do not need to change it, just clarify in the beginning.

We have added the following statements to L. 68 to clarify the definition of term TFBS in our study: *"In this study, TFBSs refer to ChIP-seq peaks"*.

Reviewer #2 (Remarks to the Author):

In this manuscript, Zhang and Huang study transcription factor binding regions which have evolved rapidly in the lineage leading to humans. This work builds on a tradition of identifying and characterising human accelerated regions. Unlike many previous papers which focused on sequences defined by evolutionary conservation (outside humans), the authors here focus on sequences defined by function. This is an interesting approach, and the work is overall interesting.

This work is related to recent work of ours, so I will sign my review: Marc Robinson-Rechavi. In a general way, I would like to point out that in Science Advances 6: eabc9863, we reported that "the TFBSs of cell types detected under selection do not necessarily evolve faster", which seems relevant to the discussion of results here.

We thank the reviewer for these kind comments and helpful feedback.

Major comments:

1 The study only considers cases of human lineage acceleration. To conclude about human evolution, it would be very helpful to contrast results to those centering on other lineages, be they other primates or mouse, for which similarly abundant ChIP-seq data is available.

In the revision, we have applied our methods to ChIP-seq data of histone modifications of both human and non-human primates, such as chimpanzee and rhesus macaque.

Vermunt et al. (2016, Nat Neurosci; 19:494-503) identified histone H3 lysine 27 acetylation (H3K27ac) enriched regions in chimpanzee and rhesus macaque brain. The H3K27ac enriched regions were predicted to be active cis-regulatory elements (CREs), We applied the group-LRT method to the predicted CREs in human, chimpanzee and rhesus macaque brain. Results revealed a slight increase in substitution rates in CREs of human and chimpanzee brain, compared to the fold of increase in substitution rate in CREs of rhesus macaque brain

Villar et al. (2015; Cell 160:554–566) identified trimethylated lysine 4 of histone H3 (H3K4me3) enriched regions and H3K27ac enriched regions in liver of 20 mammals including human and rhesus macaque. The regions were classified into active gene promoters and enhancers. Enhancers were identified by regions only enriched for H3K27ac, while promoters defined as regions containing both H3K27ac and H3K4me3. We have applied the group-LRT method to the promoters and enhancers in human and rhesus macaque. Results showed that enhancers tended to evolve faster than promoters in both species.

We have added these results in a new section in the main text (L. 286-305), methods in L. 574-593, and the results in Supplementary Table 5 (shown below).

Supplementary Table 5: GroupLRT estimates on ChIP-seq peaks of histone modifications.

Species	Regions	Tissue	Fold of increase in substitution rate(r_1/r_2)	p-value
Rhesus macaque	promoter	liver	0.95	0
Rhesus macaque	enhancer	liver	1.00	0.04
Human	promoter	liver	0.92	0
Human	enhancer	liver	0.98	0
Rhesus macaque	CRE	brain	0.97	0
Chimpanzee	CRE	brain	1.04	0
Human	CRE	brain	1.05	0

2 Acceleration might be due to positive selection, relaxed purifying selection, or biased gene conversion, as shown in many of the papers cited. This possibility has to be considered, both in the analysis and in the discussion.

In the updated manuscript, we have found NRF1 binding sites have GC-biased-gene conversion using PhastBias. Based on INSIGHT model, we have compared the selection coefficient in the seven TFBS groups and all TFBSs. The results suggest the seven TFBSs are under much weaker selection than other TFBSs ($\rho = 0.76$). We have added the following information to the results section (L. 240-259) and described the methods in the section (L. 540-556). Table 1 has been supplemented.

Table 1: Numbers of accelerated TFBSs estimated by the phylogenetic mixture model.

TFBS group	Proportion of accelerated elements ($1 - \hat{\pi}_{ub}$)	Number of elements	Number of accelerated elements	Lineage with accelerated evolution	Selection coefficient ρ	Gene conversion disparity B
Pol III binding	0.78	286	222.30	OWM & ape (M5)	0.03	0.21
BDP1	0.20	439	89.75	OWM & ape (M5)	0.02	0.24
POU5F1-NANOG binding	0.08	1341	109.90	Hominini (M2)	0.19	0.06
POU5F1	0.10	2040	204	Hominini (M2)	0.09	0
FOXP2	0.27	15881	4264.92	Ape (M4)	0.20	0
NANOG	0.26	2952	771.21	Ape (M4)	0.23	0
NRF1	0.25	1856	466.34	OWM & Ape (M5)	0.07	2.0

3 Differences in detection of selection between site categories could be due to the number of nucleotides or sites concerned, i.e. sample size. The authors should check for this, for example by down-sampling Pol III sites and verifying if selection is still detected.

We have checked the effect of sample size on the detection of selection. By downsampling the 286 Pol III binding sites to 200 or 240 binding sites, we have verified that the positive selection could still be detected in Pol III binding sites.

4 I did not understand what the authors are testing with the GO and Reactome enrichments. By testing the genes close to specific TFs, they recover the function of these TFs biased by the tissues or cell types in which the CHIP-seq was done, but we do not learn anything specific about the accelerated or selected TFBSs. I suggest to instead perform enrichment test with background all genes with a given TF, foreground all genes with that TF accelerated.

Thank you for pointing out that. In the revision, we have re-performed the enrichment analysis with genes associated with top accelerated binding sites in the seven groups, to identify the specific functions represented by the top accelerated binding sites.

We extracted the significant binding sites in each of the seven groups from the phylogenetics-based mixture model and defined them as the top accelerated binding sites. Genomic Regions Enrichment of Annotations Tool (GREAT) identified 2611 potential target genes for the top accelerated binding sites of FOXP2, 662 genes for the top accelerated binding sites of NANOG, 390 genes for the top accelerated binding sites of NRF1, 222 genes for the top accelerated binding sites of POU5F1, 163 genes for the top accelerated binding sites shared by POU5F1 and NANOG, 104 genes for the top accelerated binding sites of BDP1 and 143 genes for the top accelerated binding sites shared by Pol III TFs. Using default settings in GREAT, we built seven background gene lists for seven TFBS groups, respectively containing 9896 potential target genes for FOXP2 binding sites, 3745 genes for NANOG binding sites, 2931 genes for NRF1 binding sites, 1976 genes for POU5F1-NANOG binding sites and 478 potential target genes for POU5F1 binding sites.

After GO enrichment test and removing the redundant GO terms with high semantic similarity (0.7) and performing Bonferroni correction, we found FOXP2 top accelerated TFBSs were associated with genes functioning in artery development and regulation of transforming growth factor signaling pathway. The concatenation of top accelerated binding sites in seven TFBS groups were associated with genes playing roles in development and cell proliferation processes. In the genes associated with other accelerated TFBS groups, no pathways or biological terms were found to be significant after correction.

We have updated the enrichment plot in the revised manuscript and rewritten the sections in L. 260-285 and L. 557-573.

Figure 9: Gene ontology analysis of the genes associated with top accelerated binding sites. The dot plots show the significant GO terms for biological process of (a) genes associated with top binding sites of FOXP2 (b) genes associated with top binding sites of all seven TFBS groups. The size of circle represents the number of genes associated with top accelerated binding sites affiliated with the specific GO terms. The color of circle represents the Bonferroni-corrected p-values.

5 In the Science Advances paper cited above, we compared selection of TFBSs according to the organ or tissue where they were active. It would be very interesting to see the same here, and to compare results. This would also allow to meaningfully test subsets of TFBSs for positive selection, and maybe characterize the genes which are involved in tissue-specific adaptive acceleration.

Following the suggestions, we have applied the methods to three datasets: H3K27ac-enriched regions (predicted CREs) in human brain, H3K27ac and H3K4me3-enriched regions in human liver and human CTCF binding sites in 29 tissues/cell types. The results of the first two methods were mentioned in the response to the first comment and can be found in the revised manuscript (L. 286-305 and L. 574-593).

GroupLRT identified that human brain CREs were under accelerated evolution in human. Human enhancers evolved faster than promoters. Using group-level LRT, we only saw lower leg skin CTCF binding sites under weakly accelerated evolution in human. CTCF binding sites of brain-related tissues or cell types were found to be under selection in the Science Advance paper. However, we did not detect accelerated evolution in them. The pattern seemed similar with the Science Advances paper: “the TFBSs of cell types detected under selection do not

necessarily evolve faster". We have now cited the Science Advances paper to collaborate our references.

Statistically, the pattern might be due to a smaller sample size and lack of statistical power. Biologically, it could be true that only a small proportion of binding sites in a specific tissue were under positive selection and undergoing accelerated evolution. Given that background under negative selection, the signals of accelerated evolution might be diluted if we concatenated all binding sites.

6 I don't understand this statement: "To the best of our knowledge, our methods are the first statistical framework dedicated to infer weakly accelerated evolution." Maybe it is correct in the very specific context of non coding sequences in human evolution? Please clarify.

We have replaced this statement with "To the best of our knowledge, our methods are the first statistical framework dedicated to infer weakly accelerated evolution in non-coding regions" in L313.

Minor comments:

Line 109, "Unlikely" should be "Unlike"
Corrected.

All statements of the form x% y should be of the form x% of y, e.g. "78% of Pol III binding sites".
We have corrected these expressions in the revised manuscript.

Line 174 "more" is repeated.
Corrected.

Bonferroni takes an uppercase B.
Revised.

Reviewers' Comments:

Reviewer #1:

Remarks to the Author:

The authors have done an excellent job of addressing all of my comments and concerns from the previous review. Particularly impressive is the new analysis evaluating the newly presented statistical methods on synthetic data.

Reviewer #2:

Remarks to the Author:

In this revision, the authors have done overall a very good job of replying to the comments of both reviewers. I only have a few comments left.

1- The downsampling of Pol III sites described in the rebutal does not seem to be in the manuscript itself.

2- Most GO enrichment analyses use a less conservative FDR correction, such as Benjamini-Hochberg. It would be interesting to see the results if this were applied here.

3- In the Discussion, several new results are not taken into account, notably the scan for GC biased gene conversion, and the changes in GO enrichment.

4- The correct citation for the R language (not "program") is found at <https://intro2r.com/citing-r.html>

5- In Figure 7B, I find the black bars misleading in cases where the difference between best and second best fit is very small.

6- Figure 9 is unreadable, please use larger fonts.

7- A few new sentences are unclear and should be reformulated:

L 130 what does "as each case showed" mean?

L 134 "is slightly" should be "is only slightly"

L 255-256 this sentence is unclear: "The seven TFBS groups were under weaker selection than the collection of 161 TFBS groups"

We thank the two reviewers for reading our manuscript and offering these suggestions to improve the manuscript. We have made additional analyses, edited the manuscript, and provided a point-to-point response based on the reviewers' comments.

REVIEWERS' COMMENTS

Reviewer #1 (Remarks to the Author):

The authors have done an excellent job of addressing all of my comments and concerns from the previous review. Particularly impressive is the new analysis evaluating the newly presented statistical methods on synthetic data.

We thank the reviewer for providing these positive comments on our manuscript.

Reviewer #2 (Remarks to the Author):

In this revision, the authors have done overall a very good job of replying to the comments of both reviewers. I only have a few comments left.

We thank the reviewer for your positive comments and helpful suggestions.

1- The downsampling of Pol III sites described in the rebutal does not seem to be in the manuscript itself.

We have added the description of downsampling Pol III sites into the manuscript L.255-256:

“By downsampling the 286 Pol III binding sites to 200 or 240 binding sites, we verified that the positive selection could still be detected in Pol III binding sites.”

2- Most GO enrichment analyses use a less conservative FDR correction, such as Benjamini-Hochberg. It would be interesting to see the results if this were applied here.

In the manuscript, for GO enrichment analyses we actually used a more conservative correction – ‘Bonferroni correction’ but not a less conservative FDR correction. For all the accelerated binding sites of seven transcription factors, there were 46 terms significant after Bonferroni correction, while the Benjamin-Hochberg correction resulted in 184 significant terms. For the accelerated binding sites of FOXP2, there were only ten terms significant after the Bonferroni

correction, as compared to 69 significant terms after Benjamin-Hochberg correction. Therefore, the “less conservative” is not a concern.

3- In the Discussion, several new results are not taken into account, notably the scan for GC biased gene conversion, and the changes in GO enrichment.

In the revised manuscript, we have added the discussions of “GC biased gene conversion” and relaxed purifying selection in L.341-348 We have also added the description of GO enrichment results to the discussion in L.363-369.

4- The correct citation for the R language (not "program") is found at <https://intro2r.com/citing-r.html>

We have updated the citation for the R language (L.464).

5- In Figure 7B, I find the black bars misleading in cases where the difference between best and second best fit is very small.

For POU5F1-NANOG binding sites, the differences between the relative BICs for M2 model and M3 model is small. For POU5F1 binding sites, the difference between the relative BICs of the model M2 and M5 is small. We still took the model with the largest BIC as the best-fit model. The relative BICs are listed in Supplementary Table 2, as indicated in L.213 (shown below).

Supplementary Table 2: Relative BICs for different foreground lineages.

Model	Foreground lineage	Pol III binding	BDP1	POU5F1-NANOG binding	POU5F1	FOXP2	NANOG	NRF1
M1	Human	158.29	23.27	9.82	18.74	141.34	8.24	4.79
M2	Hominini	530.38	35.65	24.47	36.96	199.91	16.67	30.24
M3	Homininae	1060.78	40.06	24.11	23.77	211.00	10.36	33.54
M4	Hominidae	1590.02	65.20	15.94	17.96	320.93	20.07	46.08
M5	Caterrhini	1973.45	113.72	8.81	36.88	186.56	13.15	47.28
M6	Simiiformes	1110.76	58.78	2.09	20.54	28.66	0.13	6.30
M7	Haplorhini	0	0	0	0	0	0	0

6- Figure 9 is unreadable, please use larger fonts.

We have enlarged the fonts in figure 9.

7- A few new sentences are unclear and should be reformulated:
L 130 what does "as each case showed" mean?

We have revised the sentence to: “Under the first scenario, all the 200 bp binding sites in one group were assumed to be under accelerated evolution in a defined lineage as each case (1-8) showed, for example, in case 1, all the 200 bp binding sites would be under accelerated evolution in only human.”

L 134 "is slightly" should be "is only slightly"

Revised.

L 255-256 this sentence is unclear: "The seven TFBS groups were under weaker selection than the collection of 161 TFBS groups"

We have corrected this sentence: “Each of the seven TFBS groups were inferred to have a smaller fraction of sites under selection in human ρ than the collection of 161 TFBS groups ($\rho=0.76$) (Table 1). The reduced values of ρ implied weaker selection constraints in the seven TFBS groups.”